# Culture is critical in driving orangutan diet development past individual potentials

**Elliot Howard-Spink** [1,2,3] ✉, **Claudio Tennie**[4], **Tatang Mitra Setia**[5,6,7], **Deana Perawati**[5,6,7], **Carel van Schaik** [3,8,9], **Brendan Barrett**[10,11,12], **Andrew Whiten** [13] & **Caroline Schuppli** [1,3,14] ✉

Humans accumulate extensive repertoires of culturally transmitted information, reaching breadths exceeding any individual's innovation capacity (culturally dependent repertoires). It is unclear whether other animals require social learning to acquire adult-like breadths of information in the wild, including by key developmental milestones, or whether animals are capable of constructing their knowledge repertoires primarily through independent exploration. We investigated whether social learning mediates orangutans' diet-repertoire development, by translating an extensive dataset describing wild orangutans' behaviour into an empirically validated agent-based model. In this model, diets reliably developed to adult-like breadths only when simulated immatures benefited from multiple forms of social learning. Moreover, social learning was required for diets to reach adult-like breadths by the age immatures become independent from their mothers. This implies that orangutan diets constitute culturally dependent repertoires, with social learning enhancing the rate and outcomes of diet development past individual potentials. We discuss prospective avenues for researching the building of cultural repertoires in hominids and other species.

Social learning pervades the lives of humans and generates shared cultural repertoires that are indispensable for our species' survival[1]. Learning from others circumvents the costs of asocial exploration and innovation—which can be time-consuming and risky—therefore accelerating the rate at which learners can acquire fitness-maximizing knowledge and skills[2–4]. Importantly, human culture is immensely expansive[5] due to the accumulation of cultural information (cultural breadth) and the continual transformation of transmitted cultural knowledge through further elaboration and innovation (cumulative cultural evolution)[6–8]. In human culture, cumulative cultural evolution habitually produces culturally dependent traits[6,8], where the efficiency and/or complexity of socially learned behaviours, or their artefacts, exceed any one individual's innovation capacity[6,9]. Additionally, human cultures accumulate breadths of knowledge that are more expansive than any individual could generate independently[7], hereafter named culturally dependent repertoires.

Recent research has revealed widespread evidence for social learning and culture across mammals, birds and fish[10]. However, the shared

[1]Development and Evolution of Cognition Group, Max Planck Institute of Animal Behavior, Konstanz, Germany. [2]School of Biological and Behavioural Sciences, Queen Mary University of London, London, UK. [3]Department of Evolutionary Anthropology, University of Zurich, Zurich, Switzerland. [4]Working Group Early Prehistory and Quaternary Ecology, Department of Geosciences, Faculty of Science, University of Tübingen, Tübingen, Germany. [5]Department of Biology, Faculty of Biology and Agriculture, Universitas Nasional, Jakarta, Indonesia. [6]Magister of Biology, Faculty of Biology and Agriculture, Universitas Nasional, Jakarta, Indonesia. [7]Primate Research Center, Universitas Nasional, Jakarta, Indonesia. [8]Department of Evolutionary Biology & Environmental Studies, University of Zurich, Zurich, Switzerland. [9]Comparative Socioecology Group, Department for the Ecology of Animal Societies, Max Planck Institute of Animal Behavior, Konstanz, Germany. [10]Department for the Ecology of Animal Societies, Max Planck Institute of Animal Behavior, Konstanz, Germany. [11]Department of Biology, University of Konstanz, Konstanz, Germany. [12]Center for the Advanced Study of Collective Behavior, University of Konstanz, Konstanz, Germany. [13]Centre for Social Learning and Cognitive Evolution, School of Psychology and Neuroscience, University of St Andrews, St Andrews, UK. [14]Department of Comparative Cultural Psychology, Max Planck Institute for Evolutionary Anthropology, Leipzig, Germany. ✉e-mail: Elliot.howardspink@outlook.com; cschuppli@ab.mpg.de

knowledge and skills of animals typically lack many features of cultural dependence, provoking ongoing debate over whether these aspects of culture are especially pronounced in humans, particularly in comparison with other great apes (henceforth, apes)[11–14]. Compared with culturally dependent traits, little research has investigated whether animals possess culturally dependent repertoires, in part because the breadth of animals' cultural knowledge is challenging to estimate[15]. Yet, possessing broad repertoires of ecologically relevant information—including simple information surrounding the nature, location and timing of available resources—is probably fundamental for survival in wild animals. For species who have opportunities to acquire information from conspecifics, social learning theoretically may permit repertoires to expand past thresholds achievable by lone individuals[16]. There is consequently the need to develop new methodologies to assess the scope of animals' cultural knowledge, and to evaluate whether social learning enables the knowledge repertoires of wild animals to amass to culturally dependent breadths.

Many animals require knowledge of a sufficiently broad number of foods—the diet repertoire—to meet their baseline energetic and nutrient requirements for survival and reproduction; this includes during periodic turnover of available food species and phases of food scarcity[17]. Growing evidence suggests that several vertebrate species—including numerous mammals and birds—can acquire dietary information socially[18–20], particularly dietary generalists[19]. However, the majority of this evidence comes from experimental paradigms[2,21] or indirect inferences from analyses of specific foraging behaviours in the wild[22–24]. For wild animals, it has not yet been possible to empirically quantify the relative importance (if any) of different forms of social learning for developing diet repertoires with adult-like breadths, nor whether social learning is required to acquire these diets by key developmental milestones.

Given their phylogenetic proximity to humans, studies of apes can identify cultural capacities that are probably ancestral to hominins[19,25,26]. Apes' diets are vast (regularly reaching several hundred different types of food[19,27]) and potentially encompass culturally dependent breadths. Apes also exhibit extended developmental periods, affording immatures many years to learn sufficient survival-relevant information before adulthood, including knowledge and skills for foraging[26,28–30]. These developmental periods extend well beyond weaning, after which immatures must know how to source sufficient food to support the remainder of their development into adulthood. This includes the need to possess sufficiently large diet repertoires to meet the high energetic and nutrient requirements of their developing bodies and brains[29,31–33]. Observational studies on wild apes suggest that immature individuals may potentially draw on several forms of social learning during diet-repertoire development, with the primary source of these interactions being their mothers. The most prominent of these include:

(1) Following conspecifics through the home range to different areas where food is available[34,35] (broad-scale local enhancement, here named 'exposure'[36,37]).

(2) Being close to conspecifics, hence being more likely to pay attention to their precise locations in feeding patches (fine-scale local enhancement[26]) or the items conspecifics interact with (fine-scale stimulus enhancement[26]), or being more likely to experience proximity-dependent facilitation of exploration (social facilitation). We group these three proximity-dependent processes under the umbrella term 'enhancement'.

(3) Close-range observation of conspecifics' foraging behaviours ('peering'; Fig. 1a)[38–40].

Observational data suggest that these forms of social learning increase the frequency of immature apes' exploratory behaviours and target explorations towards feeding locations, food items and foraging tools[19,26,38,39,41] (Fig. 1b). However, whether these processes are required to sufficiently accelerate broad-scale diet-repertoire acquisition over

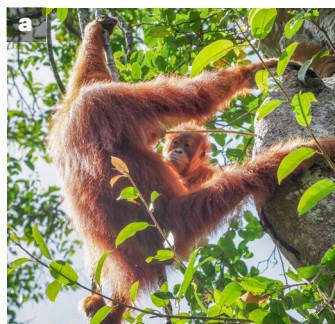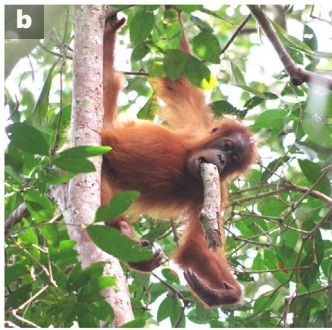

**Fig. 1 | Example peering and exploration behaviours. a**, A dependent immature (Cinnamon) is peering at her mother (Cissy), who is using a stick tool to extract termites from a nest. Peering is frequently followed by exploration of similar objects and thus can enhance learning. **b**, The same immature, Cinnamon, exploring a large stick. Photos were provided by the SUAQ Project and were taken by Guilhem Duvot.

apes' extended developmental periods has defied empirical testing. As apes encounter many thousands of food items during development, it has not been possible to rule out whether adult-like diets could emerge without social learning's influence on exploration, as less frequent exploration may be sufficient given the immense number of learning opportunities afforded to immatures. Quantitatively evaluating the effects of social learning on long-term diet-repertoire development, in relation to animals' key developmental milestones, will reveal the extent to which wild animals are dependent on cultural processes to acquire fundamental information for survival.

Unlike African apes, orangutans are largely solitary after becoming independent from their mothers (aside from mothers' own close association with younger, dependent offspring). Immature Sumatran orangutans (*Pongo abelii*) are weaned at around 7.5 years[30] but often continue ranging with their mothers until the age of 8–9 years, regularly feeding in close proximity before starting to range independently[35]. Additional individuals form transient associations with mother–infant dyads, providing further sources of information over shorter timescales[27,39]. Overall, opportunities for social learning are disproportionately skewed towards earlier years of orangutans' lives, which immatures make use of following initial motor and cognitive development but before the onset of independence, after which further opportunities are relatively rare. Observational data show that by the onset of independence, orangutans have accumulated adult-like diet repertoires[27], and the breadth of these knowledge repertoires is probably crucial to support immatures' transition to energetic independence. This developmental milestone therefore presents a so-far-unexploited opportunity to assess whether orangutans are dependent on different forms of social learning to acquire adequately large diet repertoires sufficiently quickly to become independent foragers. However, this question cannot be answered from observational data alone, as documenting all social-learning opportunities afforded to different immatures, over periods of years, is logistically unfeasible.

## Social learning experiments in-silico

We formulated an empirically-informed agent-based model (ABM) of orangutan diet-repertoire development that was entirely calibrated using over 12 years of behavioural data collected from wild Sumatran orangutans. We directly translated data from wild individuals into model variables that predict the likelihood that simulated immatures (that is, agents) would encounter different foods, enter social states that reflect association and social behaviours (for example, close association proximity or peering), and consequently explore food items to facilitate broad-scale diet learning (thus, simulating the emergent process of orangutans' long-term diet learning through localized interactions, up to the end of an orangutan's immature period at 15 years; Fig. 2).

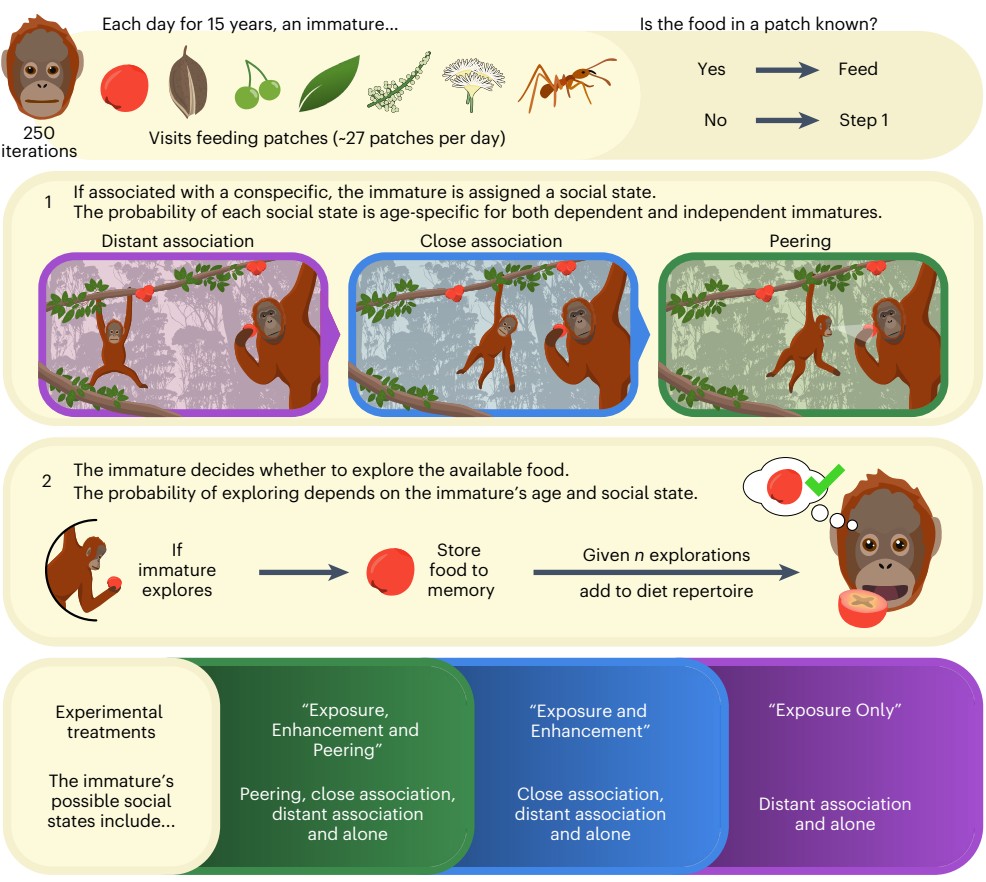

**Fig. 2 | ABM structure and experimental treatments.** We simulated the developmental trajectory of one immature per ABM iteration. Simulated immatures had no initial knowledge of any available food items. Each day, the ABM generated a series of feeding patches (sampled from a Poisson distribution calibrated to the feeding opportunities of mothers, with a mean of 27 patches), and simulated immatures visited these feeding patches sequentially. Each patch contained one type of food item. If the food item was known to the simulated immature, they were marked as 'feeding'. Otherwise, the simulated immature either explored the food or moved on to the next patch without exploring. To calculate the likelihood of exploration, we assigned a social state to the immature that related to the available forms of social learning. We first calculated the probability that an immature would be alone or associated with a conspecific. If associated with a conspecific, immatures were assigned to either distant association (thus only subject to exposure) or close association (therefore subject to enhancement). Immatures in close association could also then engage in peering at a conspecific (therefore potentially supplementing learning through observation of social partners). The probability of being in each social state was dependent on a simulated immature's age. The likelihood of exploring an unknown food was dependent on an immature's age and social state (and for all combinations of age and social state, there was still a non-zero probability that immatures chose not to explore food items). If a simulated immature had explored a food item enough times to learn how to process it correctly for consumption, the food item was added to their diet repertoire. The required number of explorations to learn how to consume a food item was scaled to its processing complexity. All model parameters were calibrated using data from Suaq (Methods).

We then validated whether, through exclusively using data-driven inputs, our ABM successfully captured the dynamics of orangutan diet development using observational data from wild immatures.

Following ABM validation, we experimentally withheld different forms of social learning from cohorts of simulated immatures to assess their potential impact on long-term diet development, first via removing observational social learning through peering, and then more generalized processes of enhancement. We then quantified whether (1) adult-like diets could still emerge by key developmental milestones without these forms of social learning, including solely through exposure to thousands of learning opportunities with different food items during immaturity, and if not (2) the relative importance of different forms of social learning for diet development, estimated via reductions in diet breadth when forms of social learning were withheld from the simulation.

## Results

A summary of the ABM structure is provided in Fig. 2 (see Methods for a step-by-step description of the ABM process). All data used to calibrate the ABM variables were collected from wild Sumatran orangutans living in the Suaq Balimbing research area (2007–2019), including 2,676 focal follows on 132 individuals, totalling 22,547 observation hours.

### Wild adult diet-repertoire size

We identified 262 distinct food items (from 116 identifiable plant species and nine higher-level taxonomic food groups, such as termites). To estimate adult diet-repertoire size, we used the Michaelis–Menten equation to estimate the plateau for cumulative diet-repertoire size with increasing sampling effort for each adult: 248 food items ($N_{Adults} = 95$; $N_{Follows} = 1620$; $N_{Scans} = 402{,}082$; Supplementary Table 2). Wild orangutans' diet repertoires vary between individuals[27,42], and orangutans learn a small number of foraging behaviours after becoming independent[43]. We thus compared simulated immatures' diet repertoires to a reduced threshold (adult-like repertoire), which is 90% of the adult repertoire size (223 food items).

### Exposure to food items in the wild

To estimate the number of food items that wild immatures are exposed to per day, we calculated the number of feeding patches mothers (and therefore their dependent immatures) visited during daily focal follows

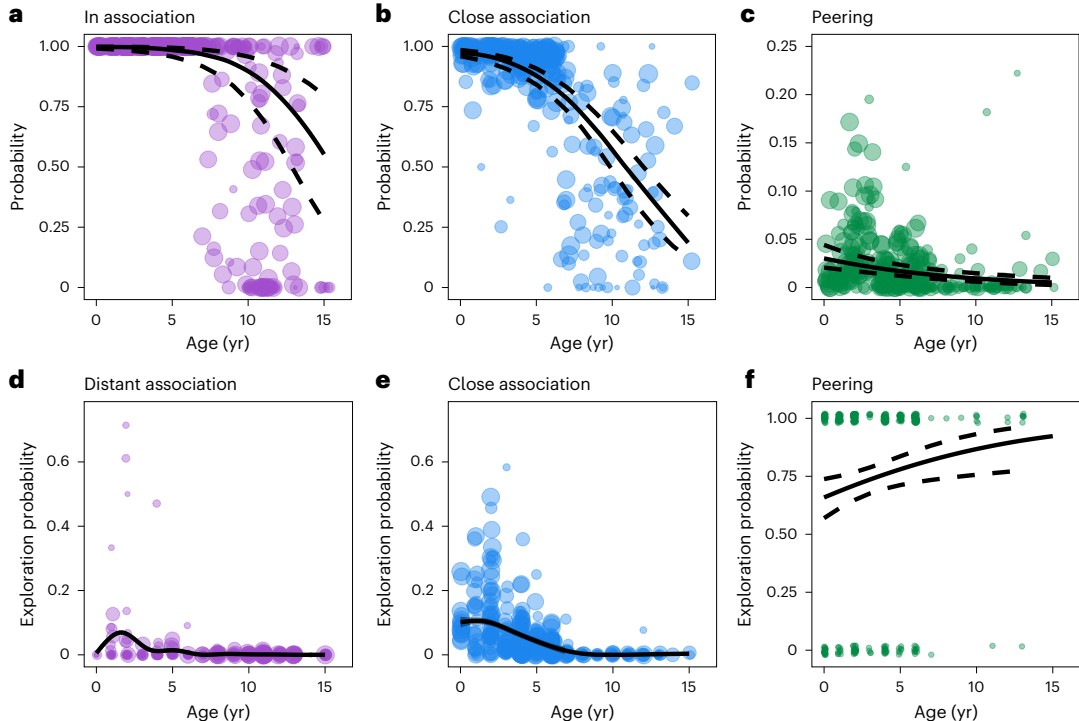

**Fig. 3 | Estimation of wild orangutan social states and exploration across development. a–f**, The probability of being associated with a conspecific (**a**); the probability of being in close association if associated with a conspecific (**b**); the probability of peering if in close association (**c**); the probability of exploring when in distant association (**d**); the probability of exploring when in close association (**e**); and the probability of exploring following peering (**f**). For **d** and **e**, the smoothed trend lines describe GAMM predictions for all data points. For all other plots, the solid lines are mean predictions from GLMMs, and the dashed lines indicate the 95% CI for the regressions. The mean values for each coefficient, at each developmental age, can be found in Supplementary Table 1. Outputs of statistical tests confirming the significance of the regression variables can be found in Supplementary Tables 2–4 as well as in Results.

(follows at least 10 hours long to capture all feeding behaviours per day). On average, mothers visit 27 feeding patches per day ($N_{\text{Mothers}} = 13$; $N_{\text{Follows}} = 252$; Supplementary Table 2). For our ABM, the number of feeding patches encountered per day was generated using a Poisson distribution with a mean of 27 patches. The content of each feeding patch was determined by the relative rates at which different food items are encountered at Suaq ($N = 262$ possible food types; see the associated data for the probability of encountering different foods).

### Social states across development

We calibrated the probability that simulated immatures would be in each social state across development using behavioural and association data from wild individuals. These social states were (1) alone; (2) distant association; (3) close association, therefore benefiting from enhancement; and (4) peering, therefore supplementing learning through observation of social partners.

To calibrate the probability that simulated immatures were either alone or associated with a conspecific in a given feeding patch, we modelled the age-dependent association rates of wild immatures ($N_{\text{Immatures}} = 30$; $N_{\text{Follows}} = 362$; $N_{\text{Scans}} = 99,264$). Age was significantly negatively correlated with association rate (quasibinomial generalized linear mixed model (GLMM): $t_{331} = -6.7$; $P < 0.001$; age $= -0.4$; 95% confidence interval (CI), ($-0.51$, $-0.28$)). Wild immatures were in near-constant association with at least one conspecific (including their mothers) up to the age of 7 to 9 years (in >99% of scans from one to 3 years and in >97% of scans from 4 to 7 years), after which they spent progressively more time ranging alone (Fig. 3a). This time period (up to 7 to 9 years) matches the dependency phase, when immatures spend all of their time ranging with their mothers, with additional individuals sometimes associating with the mother–infant dyad for limited time periods. We directly translated the model's prediction of association rates across development into the probability

of simulated immatures being associated with a conspecific in a given feeding patch (Supplementary Table 1). Simulated immatures who were not associated with conspecifics in a given feeding patch were marked as alone.

For simulated immatures who were associated with at least one conspecific at a given feeding patch, we modelled the likelihood that they would be in a state of close association (within 10 m) or distant association (10–50 m) with another individual. When they were associated with a conspecific, the likelihood that wild immatures were in close association reduced significantly with age ($N_{\text{Immatures}} = 28$; $N_{\text{Follows}} = 323$; $N_{\text{Scans}} = 78,478$; quasibinomial GLMM: $t_{294} = -11.2$; $P < 0.001$; age $= -0.339$; 95% CI, ($-0.4$, $-0.28$)). The probability of being in close association steadily decreased with age between birth and 6 years (97.5% at birth; 83.8% at 6 years), before decreasing more rapidly over subsequent years (Fig. 3b and Supplementary Table 1). By the final year of immaturity (14 years), wild individuals spent 25.5% of their time in close association with at least one conspecific. We used these values to calibrate the probability of a simulated immature being in close association across development, when associated with at least one conspecific at a feeding patch. All simulated immatures who were associated with at least one conspecific in a feeding patch, but not in close association, were marked as being in distant association.

Immature orangutans regularly peer at conspecifics, and given that peering is frequently directed at older individuals (usually mothers[44]) as well as rare and skill-intense behaviours[38], peering occurs in contexts where learning can be expected. We modelled wild immatures' peering rates when in close association across development ($N_{\text{Immatures}} = 28$; $N_{\text{Follows}} = 311$; $N_{\text{Scans}} = 66,501$). We found a significant negative relationship between the probability of peering at a conspecific and age when in close association (quasibinomial GLMM: $t_{282} = -4.52$; $P < 0.001$; age $= -0.12$; 95% CI, ($-0.17$, $-0.07$); Fig. 3e and Supplementary Table 1). We directly translated this relationship into

the age-dependent probability that simulated immatures would be in the peering state when in close association in a given feeding patch.

### Exploration in each social state

Immature orangutans have a generalized drive to explore, which is dependent on their age and can be further enhanced by environmental factors and their social state[27,35,38,39,44–46]. We therefore calibrated exploration rates to both age and social state using behavioural data from wild immatures.

Exploration rates were significantly higher when wild immatures were in close association than in distant association ($N_{Immatures}$ = 30; $N_{Follows}$ = 364; $N_{Scans}$ = 81,153; binomial generalized additive mixed model (GAMM) with close and distant association as categorical factors; $\chi^2$(effective degrees of freedom = 9.84) = 1,754.25, $P$ < 0.001; Fig. 3c and Supplementary Table 1). In both conditions, exploration rates peaked during the first few years of life (between 1 and 3 years of age), before decreasing over subsequent ages (Fig. 3d,e and Supplementary Table 1). Across development, we set the mean exploration rates of simulated immatures in close and distant association to be equal to the probability of wild immatures exploring in either association condition. As wild immatures spend virtually all of their time in association up to 7 to 9 years (see above), we could not estimate an exploration rate for immatures who are alone at these ages. We therefore set the exploration rate when simulated immatures were alone to be the same as when in distant association.

To determine whether peering influences exploration, we compared the probability that wild immatures explored food items in the hour before and the hour after peering. We found a significant difference in the rate of exploration in the hour after peering (probability of exploration, 67.7%; 95% CI, (59.8%, 74.7%); $N_{Immatures}$ = 13; $N_{Follows}$ = 86; $N_{PeeringEvents}$ = 238; data refined to peering events where food items had not been explored before peering), compared with the hour before peering events (33.5%; 95% CI, (26%, 42%); $N_{Immatures}$ = 14; $N_{Follows}$ = 101; $N_{PeeringEvents}$ = 367), indicating that peering is associated with significantly higher exploration rates. The exact relationship between peering and exploration was age-dependent, with peering being more likely to be followed by exploration at older ages ($N_{Immatures}$ = 14; $N_{Follows}$ = 101; $N_{PeeringEvents}$ = 367; binomial GLMM: $Z$ = 2.57; $P$ = 0.0102; age = 0.12; 95% CI, (0.03, 0.22); Fig. 3f and Supplementary Table 1). To ensure that this result was not an artefact of small sample sizes at ages above 6 years old, we replicated our analysis using data between the ages of birth and 6 years, which found a similar effect ($N_{Immatures}$ = 8; $N_{Follows}$ = 90; $N_{PeeringEvents}$ = 348; binomial GLMM: $Z$ = 2.39; $P$ = 0.017; age = 0.14; 95% CI, (0.03, 0.25)). These results confirm that, while peering behaviours are performed less frequently at older ages (especially once orangutans establish independence), instances of peering at older ages are more likely to be followed by exploration of food items. We used the predicted probability of exploring after peering at each age to calibrate the likelihood that simulated immatures would explore food items when in the peering state.

### Simulated diet-repertoire development

To assess whether social learning influences the size of emerging diet repertoires by key developmental milestones, we ran three sets of simulations where simulated immatures had access to different combinations of social learning (250 iterations per treatment):

(1) Exposure, Enhancement and Peering: in a given feeding patch, simulated immatures could be allocated to any of the four social states, including alone or distant association (thus benefiting from exposure), close association (thus also benefiting from enhancement) or peering (thus benefiting from upregulated exploration following social observation).

(2) Exposure and Enhancement: an identical protocol, but simulated immatures could not engage in peering.

(3) Exposure Only: an identical protocol, but simulated immatures could not engage in peering or enter close association with a conspecific at a feeding patch.

Across simulations, we compared the mean repertoire sizes at two key milestones: (1) the onset of independence from the mother (that is, the maximum age of independence, 9 years, after which immatures are responsible for their own energy intake through independent foraging) and (2) the end of immaturity (that is, the age of females' first reproduction, 15 years).

By the end of immaturity, simulated immatures ($N$ = 750) had visited an average of 147,807 feeding patches (s.d. = 382). Of the simulated immatures who could enter close association with conspecifics and engage in peering (Exposure, Enhancement and Peering), 78.4% of individuals developed adult-like diet repertoires by the end of immaturity (196/250 individuals; Fig. 4). The mean age at which these simulated immatures developed adult-like diet repertoires was 7.5 years (95% CI, (7.25, 7.84 years)). We found no significant difference between the mean age at which adult-like repertoires emerged in these simulated immatures and the age at which adult-like repertoires develop in wild individuals (that is, mean age of independence at Suaq, 8.2 years; 95% CI, (7.7, 8.6 years); $N_{Wild}$ = 8; Linear Model: $t_{202}$ = 0.85; $P$ = 0.398; wild = 0.63; 95% CI, (−0.84, 2.11)). Across all simulated immatures in this treatment, the mean repertoire size at the onset of independence reached the adult-like threshold (224 food items; 95% CI, (221, 226); adult-like threshold, 223 food items; total foods available in the environment, 262). These results confirm that our model captures both the learning trajectory and outcomes observed in wild immatures, thus validating our ABM's real-world relevance.

When simulated immatures were prevented from peering (Exposure and Enhancement), the rate of diet development was initially similar to before; however, diet development began to slow across the dependency period, leading to the development of smaller diets (Fig. 4). When peering and close association states were removed (Exposure Only), diet learning was initially slow, followed by rapid expansion and then a subsequent plateau in development (this mirrored the exploration rates in distant association over development; Fig. 3d).

Slower diet development without social learning was reflected in significantly smaller diets at the onset of independence (compared with Exposure, Enhancement and Peering: Exposure and Enhancement: $Z$ = −8.3; $P$ < 0.001; −11 food items; 95% CI on diet reduction, (−13, −9 foods); Exposure Only: $Z$ = −58.3; $P$ < 0.001; −72 food items; 95% CI on diet reduction, (−74, −69 foods)) and at the end of immaturity (Exposure, Enhancement and Peering, 226 food items; in comparison, Exposure and Enhancement: $Z$ = −9.25; $P$ < 0.001; −12 food items; 95% CI on diet reduction, (−15, −9 foods); Enhancement Only: $Z$ = −58.2; $P$ < 0.001; −72 food items; 95% CI on diet reduction, (−70, −75 foods)).

Overall, only 1.6% of simulated immatures (4/250 individuals) developed adult-like repertoires by the end of immaturity when they were prevented from peering (Exposure and Enhancement). When simulated immatures were prevented from both peering and entering close association (Exposure Only), no immatures developed adult-like diet repertoires by the end of immaturity. Given that immatures are virtually always accompanied by their mothers in the wild, we could not remove the effects of exposure on diet development in a data-driven manner. Instead, we provide a general 'Zone of No Exposure' to visualize the range of possible developmental trajectories if exposure was limited (Fig. 4). We provide tentative internal resolution for developmental trajectories within this zone using step-wise reductions in the mean number of foods simulated immatures encountered each day (Discussion and Extended Data Fig. 1).

### Discussion

Humans habitually accelerate their information acquisition through social learning and accumulate extensive, culturally dependent repertoires of socially transmitted information. We demonstrate that these

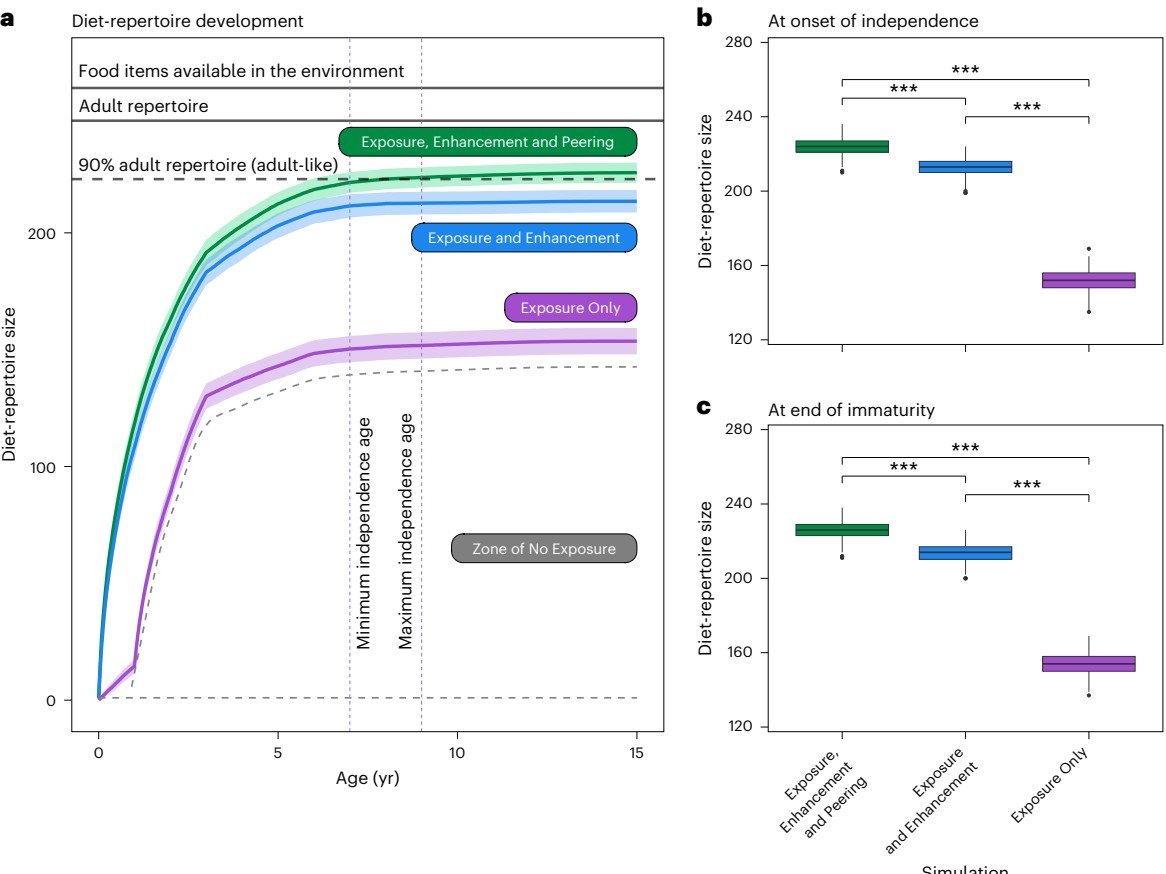

**Fig. 4 | The effects of different forms of social learning on diet-repertoire development and size. a**, Diet-repertoire development over simulated time. The horizontal lines indicate the total number of food items in the simulated environment, the estimated size of the adult diet repertoire at Suaq and the adult-like repertoire (90% of the estimated adult diet repertoire). The minimum and maximum ages for the onset of independence are indicated by vertical dashed purple lines. The Zone of No Exposure (contained between grey dashed lines) indicates all possible outcomes if we also modified the rate at which simulated immature orangutans were exposed to different food items (thus, a theoretical baseline control for the removal of peering, enhancement and exposure; see Extended Data Fig. 1 for simulations of development within this zone). For each treatment, a weighted curve indicates the mean repertoire size across development, and the shaded region around each curve indicates ±1 standard deviation of the mean. **b**, Diet-repertoire size of simulated immatures at the maximum age for the onset of independence (nine years). **c**, Diet-repertoire size at the end of immaturity (15 years). For **b** and **c**, N = 250 simulations per treatment. Experimental treatments are indicated in different colours. Three asterisks (***) indicate a significant difference between treatments of P < 0.001, with significance determined by a Poisson GLM (two-sided tests; all means compared in the same model; see 'Simulated diet-repertoire development'). In the box plots, the centre lines indicate the medians, the top and bottom boundaries of boxes indicate the upper and lower quartiles, the whiskers extend to 1.5× the interquartile range, and the points indicate outliers.

effects exist within the culture of a non-human species: the Sumatran orangutan. Our simulation of orangutan diet-repertoire development was entirely calibrated using long-term data from the wild. When this model included multiple forms of social learning that previous studies have suggested are important for orangutan diet development[38,39,44], it simulated diet-repertoire development that accurately matched the outcomes and timing observed in wild individuals. This validation ensured that, using exclusively data-driven inputs, our model produced a trajectory of diet development comparable to that of wild orangutans[47]. We then evaluated the long-term effects of removing forms of social learning on this developmental trajectory.

The removal of even one form of social learning from our ABM— peering—significantly decelerated diet-repertoire development and prevented the reliable emergence of adult-like diet repertoires during immaturity. This effect was further exacerbated when both peering and enhancement were removed, where no simulated immatures developed adult-like diet repertoires (Fig. 4). Despite simulated immatures visiting approximately 148,000 feeding patches during their immature period, these highly recurrent opportunities to encounter different food items were not sufficient for broad-scale diet development without the additional effects of peering and enhancement on immatures'

exploration. Moreover, diets reliably developed to adult-like breadths by the onset of independence (a key milestone for wild individuals) only when both peering and enhancement were included within the ABM. Our findings therefore support the hypotheses that social learning accelerates orangutan diet-repertoire development and ultimately produces adult diets that are broader than any one individual could construct independently (or even when orangutans could still benefit from broad-scale exposure). Orangutan diets therefore constitute a likely example of a culturally dependent repertoire in a non-human species.

The removal of enhancement effects on immatures' exploration (once peering had also been removed) had a much greater impact on repertoire development than the removal of peering alone. Enhancement processes are thought to lead to broad-level information acquisition, whereas observational social learning probably serves to acquire detailed information[4,38]. As many of the food items in orangutan diets involve relatively simple preingestive processing behaviours[44], enhancement is probably sufficient to acquire this (largest) proportion of orangutans' diet repertoires. The removal of peering behaviours had a smaller but significant effect on repertoire size in our simulation. Observational social learning through peering potentially plays an important role in enabling orangutans to acquire more complex

food-processing behaviours (for example, those with a greater number of processing steps). This hypothesis is supported by wild immatures peering more frequently at food items that are rare and require multi-step processing behaviours[38]. We did not program simulated immatures to peer more frequently at complex foods, as we lacked sufficient data to reliably estimate exploration after peering if foods were categorized by complexity level. We also confined the effect of peering to its effect on exploration rates (as observed in the wild), without consideration of additional information that individuals may acquire about food-processing behaviours during social observation (not quantifiable using the available data). Consequently, our model offers conservative estimates for the importance of peering for incorporating complex foods into an individual's diet.

Prior studies of cultural dependence in animals have traditionally focused on whether particular cultural traits are too complex for naive individuals to innovate from scratch[11–14], or whether, through cumulative cultural evolution, traits reach a level of efficiency that individuals cannot reach through iterative practice alone[48–50]. Whether individual animals possess a breadth of cultural knowledge that exceeds individual innovation—and thus, culturally dependent repertoires—has by contrast been a neglected question. However, the breadth of cultural knowledge possessed by an individual probably has important effects on their fitness, particularly in the foraging context, where foraging-related information can modulate the total number of calories and nutrients acquired during daily activity[17]. While the acquisition of highly technical skills may also help overcome such issues—as in the case of sophisticated tool-use behaviours used for foraging—the fitness consequences of not learning specific technical skills may prove smaller than the fitness costs of missing out on large proportions of available food species in an individual's environment. Even for orangutans, highly complex foraging behaviours—including those requiring tools—occupy a fraction of their diet and feeding time[51].

Diet breadth in wild apes, including to the extent of cultural dependence, probably has the potential to modulate considerable aspects of an individual's fitness. Future research should aim to quantify these fitness effects in more detail, such as through estimating the possible costs of individual learning used to compensate for reduced diet breadths, or through evaluating the consequences of reduced diet size for individuals' daily calorie intake, especially following the onset of independence. During this time, immatures are at specific risk, as they face considerable energetic demands to support their own development while still lacking adult-level proficiency in foraging-related skills[32,33]. Building on our results, further understanding the fitness consequences of reduced diet breadth for calorie and nutrient gain—and its consequences for survival and reproduction—will be informative for planning effective reintroduction strategies used for orphaned wild orangutans. Currently, such reintroduction programmes teach immatures survival-relevant knowledge and skills before reintroduction to the wild through 'forest schools'[52,53]. These programmes should continue to consider the importance of social learning for orangutan diet development, including the necessary sizes and compositions of taught repertoires that maximize survival likelihood post-reintroduction.

The existence of culturally dependent diet repertoires in orangutans raises the question of whether orangutan diets represent heritable and stable sets of cultural knowledge[54], produced by the exploration and learning of many individuals across generations. The diet repertoires of immature orangutans closely reflect those of their mothers, suggesting that repertoires are inherited across generations[27,44]. However, immatures also make use of social interactions with non-maternal adults to enhance learning[38,39], and upon reaching adulthood, males ranging into new areas employ peering behaviours to acquire information about previously unencountered foods[43]. These findings suggest that diet repertoires can be additionally shaped by oblique and horizontal information transmission. Further research into the stability and heritability of orangutan diets will hold

the key for understanding the fidelity with which they are culturally transmitted across generations. Moreover, along with our findings, this research may further clarify whether variation in diets observed between orangutan populations[55]—and between populations of other apes—is the product of intergenerational cultural inheritance.

Like all ABMs, our model is a careful simplification of reality that permits characterization of how localized behaviours influence a complex system[47,56]. In calibrating and validating our ABM using long-term real-world data, we faced some limitations, which required us to construct our model in a way that probably led to conservative estimates of the effects of social learning on diet development (also see Supplementary Information section 1):

First, we did not experimentally remove exposure as a form of social learning, despite its likely importance for diet learning[37]. Wild dependent immatures cannot survive without mothers, as they rely on energetic supplements through nursing while they learn vital information and skills for survival[30,35,57]. Dependent immatures also benefit from locomotor support when being carried through sections of the canopy[57,58] and the provision of suitable sleeping environments through nest sharing[35]. As dependent immatures are never without their mother, we could not estimate how frequently dependent immatures would be able to find, explore and consume food items in the environment without being guided to feeding patches. We argue that without exposure, diets are likely to be significantly smaller than those we have estimated (see 'Zone of No Exposure' in Fig. 4a), and we outline some tentative results for how variable exposure rates could influence diet development in Extended Data Fig. 1. However, we predict that without the crucial maternal effects that accompany exposure, dependent immatures are unlikely to survive even for short periods. Future research should aim to further quantify the importance of exposure for diet learning (using ethically informed approaches, such as testing how social environments influence novel food acceptance in captivity[37]).

Second, we focused on the importance of two forms of social learning: enhancement and observational social learning through peering. These forms of social learning are observed frequently in the wild; however, we acknowledge that additional forms of social learning are available to orangutans, such as food solicitations[59,60]. Food sharing is performed at low rates by mother–offspring dyads up to the onset of independence[60], usually for difficult-to-process food items. Given that these behaviours are comparatively rare, we could not realistically estimate whether they influence long-term diet-repertoire development. It is possible that such episodes facilitate fully matured diet-repertoire development, and their relative contribution should be assessed where possible.

Third, smaller diet repertoires may increase hunger and hence produce a greater motivation to find new food items through independent exploration. We did not program simulated immatures to upregulate their exploration behaviours if diet repertoires were too small, as the dynamics of this could not be estimated from wild data. Mature rehabilitant orangutans (who have smaller diets) can sometimes upregulate exploration following release to the wild[61]; however, this frequent exploration also comes with detectable risks of consuming one of the numerous toxic species that occur in orangutans' natural habitats[52,61,62]. Fatal feeding errors have been reported during the reintroduction of some primate species to novel habitats[52,61,63]; therefore, upregulation of unguided, independent exploration may be risky for diet expansion, with possibly devastating fitness consequences.

We conclude that orangutans' diets are culturally dependent knowledge repertoires possessed by a non-human species, and that orangutans' diets develop sufficiently quickly only through social learning. The ability to construct culturally dependent breadths of knowledge may have been shared by the most recent common ancestor of humans and the other great apes around 13–15 million years ago[64]. Our findings also provide evidence that the cultural knowledge possessed by the common ancestors of hominids—and species across

the hominin lineage—may have included basic subsistence knowledge used for day-to-day decision-making and was potentially broader than surviving artefacts suggest. Further research is required to understand whether wild individuals of other ape and non-ape species possess culturally dependent knowledge repertoires, with species whose diet repertoires vary in size and breadth between populations offering suitable candidates for validation. Characterizing broad-scale development of other types of socially acquired, fitness-relevant knowledge may unveil additional examples of culturally dependent repertoires, such as for sociocommunicative behaviours[65–67], navigation[68] or construction behaviours such as nest building[69]. This research may reveal that cultural processes modulate the extended breadth of knowledge possessed by many wild animals, which could consequently shape their evolution.

## Methods

### Ethics statement

Approval for data collection protocols at Suaq was provided by the Indonesian State Ministry for Research and Technology (RISTEKDIKTI) and the National Research and Innovation Agency (BRIN). The reporting of this study meets ARRIVE guidelines.

### ABM structure

We simulated the developmental trajectory of orangutan diet repertoires day by day for the duration of their immature period, up to the approximate age of first reproduction (15 years). Simulated immatures initially had no known food items in their diet repertoire. Each day, we generated the number of feeding patches a simulated immature would visit by sampling from a Poisson distribution, with a mean equal to the mean number of patches orangutan mothers at Suaq visit per day (27 feeding patches). For each feeding patch, we generated a single type of food item on the basis of the rates at which different foods were encountered at Suaq. Additionally, for each feeding patch, a simulated immature was assigned one of four social states: (1) alone, (2) distant association, (3) close association and (4) peering. These four states were chosen because wild immatures exhibit more frequent exploratory behaviours when closely associated with conspecifics[44] and after peering[38]. In the wild, the probabilities of being in close association with a conspecific and engaging in peering, and the resultant likelihood of exploring food items in both of these contexts, all change over the course of development[27,44,58]. We therefore calibrated the probability that simulated immatures explored the food item in a feeding patch by both their age and their social state, using estimates from long-term data on wild orangutans. If a simulated immature performed an exploration behaviour, this exploration was logged in a list of explorations performed by the immature across their lifetime. At the point when a particular food item had been explored enough times to meet a minimum threshold for learning, this food item was added to the simulated immature's diet repertoire (with the number of required explorations being scaled to the complexity of the food item's associated processing behaviour). If a simulated immature encountered a food item that was already present in their diet repertoire (and therefore known), their behaviour was marked as 'feeding'.

Our simulated immatures did not move around a simulated space (that is, our ABM is spatially implicit) but instead were presented with food items and social environments that reflect the opportunities presented to wild orangutans (akin to individuals being positioned in front of a 'conveyor belt' of food patches, which they are exposed to in turn, alongside a generated social state for each patch). This allowed us to calibrate our model precisely to reflect the opportunities afforded to immatures in the wild and circumvent unnecessary error introduced through estimations of movement rates across simulated environments. All steps to verify that our model operates following programmed protocols can be found in this Article's associated code.

All ABM coefficients were calibrated using data from wild individuals. Further information about the statistical models we used to analyse data from wild individuals—including the rationale behind the choices of our models and their structure—can be found in Supplementary Information section 2 (alongside all model summaries and all resulting coefficients for the ABM; Supplementary Tables 2–5). The experimental treatments we applied to our ABM are outlined in Results, and for each experimental treatment, we ran 250 iterations (a compromise between large sample size and available computational power). When analysing data from wild individuals, we used all available data where possible (unless otherwise specified for a particular analysis), and these sample sizes are comparable to or larger than those of similar studies[27,29,30,38,39,42–45,57–60,69].

### Study site and long-term data collection

We calibrated our model using data on wild orangutan behaviours collected at Suaq Balimbing (South Aceh, Indonesia). Since 1994, data have been collected at Suaq through daily focal follows, where observers collect behavioural and association data using instantaneous scan sampling at 2-minute intervals. All-occurrence sampling of key behaviours is also performed during this time frame—for example, on peering behaviours (the full protocol for data collection can be found at https://www.ab.mpg.de/571325/standarddatacollectionrules_suaq_detailed_jan204.pdf). We sampled data collected between 2007 and 2019, including 2,676 follows on 132 individuals. When estimating parameters for our ABM, we used specific subsets of these data on the basis of the relevancy of the ages and social classes of focal individuals and, where necessary, the length of focal follows.

All individuals included in this study are well known and form part of long-term data collection at Suaq. The ages of immature individuals were estimated on the basis of known births or physical characteristics at their first encounter. Immature individuals are classed as 'dependent immatures' up until the age they are observed ranging independently from their mothers for at least two consecutive follows. After this age, immatures are classed as 'independent'. Immatures reach independence at Suaq at between 7 and 9 years of age (mean, 8.1 years; $N = 8$; maximum age estimated, 9 years). We used the maximum age—9 years—as the threshold age for the onset of independence, thus offering a generous baseline for the latest reasonable age for adult-like diets to develop in the wild. Following the onset of independence, associations with mothers and other individuals become progressively less frequent[35]. Opportunities to benefit from other individuals' diet knowledge thus become rarer, and immatures must increasingly rely on their own foraging knowledge for survival. Individuals are classified as adults once they reach the average age at which females first reproduce (15 years[30]; females who reproduced before this age are classed as adults from the age they give birth to their first offspring).

Whenever the focal individual was feeding or exploring a food item at a 2-minute scan, the food item was noted. Food items are recorded at Suaq as a combination of the consumed species and, where relevant, the part of the species being consumed. For plants, food items are differentiated according to the organ(s) being consumed (for example, 'Leaf', 'Fruit', 'Pith', 'Bark', 'Flower', 'Seeds' or the general vegetative material of the plant, 'Veg'). For insects, we recorded the specific type of insect on the level of the family or clade (for example, all ants are coded as 'Semut') but did not differentiate on the parts of the body given they were often consumed whole. Food items were classified according to the complexity of the behaviour required to process the item before ingestion[29] (on a ranked scale from 0 to 5; see below), including all steps to acquire edible material and dispose of waste. Items eaten whole (for example, many types of leaves) are marked at complexity level 0. Each additional processing step (for example, peeling or spitting out inedible material) increases the complexity score by one. Behaviours that involve the use of tools are marked at the highest complexity score, 5.

### Adult diet-repertoire size

Because estimates of diet-repertoire size are highly dependent on observation time[27], we estimated the total size of the diet repertoire of adults at Suaq by modelling the cumulative number of different food items consumed by specific adults as a function of sampling effort (measured as the number of behaviour scans). To estimate the total adult repertoire size, we fit a nonlinear mixed-effect model using the Michaelis–Menten equation[70], with individual identity as a random intercept (the model was fit using the nlme package[71] v.3.1.168). Using the Michaelis–Menten equation permitted us to estimate the asymptote in repertoire size as a function of sample effort ($V_{max}$). This asymptote was taken to be representative of the total adult repertoire size.

### Feeding patches

To estimate the mean number of feeding patches that wild dependent immatures visit in a day, we used the number of feeding patches visited by their mothers. Dependent immatures consistently follow mothers through their home range[57,58] and are therefore exposed to a similar number of feeding patches—and array of different food items—per day. The number of feeding patches visited per day can be reliably estimated for mothers, as immatures may not feed if a food is not yet within their diet repertoire (that is, if it is unknown). We inferred the number of feeding patches that mothers visited via counting the continuous periods in which mothers consumed specific food items, as recorded in the focal follow data. We did not pay attention to breaks from feeding introduced by other behaviours that took place at feeding patches (for example, nursing or resting). However, if a mother began eating a different food item, this was classed as entering a new feeding patch. Our estimates include the possibility that mothers visit multiple feeding patches across a given day that contain the same food.

To model the average number of feeding patches visited by mothers, we constructed a Poisson GLMM to model the mean number of feeding patches encountered per follow, with focal ID as a random intercept.

To model the content of feeding patches, we estimated the probability of different food items being encountered across all follows for all adult orangutans. Including data from all adults ensured that our sample size was sufficiently large to yield reliable estimates of the probability of encountering all food items at Suaq, including foods that are rarely eaten. For each food item, we counted the number of follows (and therefore days) this food item was available for consumption (its food item frequency). By dividing each food item frequency by the sum total of all frequencies, we generated a discrete probability distribution for the probability of encountering different food items, which is proportional to how frequently each food is observed being consumed in the focal follow data (see the associated data for the full list of food items and their probabilities of being encountered). Each feeding patch contained a single type of food item; while different foods can occur in close physical proximity in the wild (for example, leaves and fruits from the same tree may be considered both in the same patch), restricting each patch to one food item closely resembled how we estimated the number of different feeding patches mothers encountered per day (where swapping between foods in close proximity would have been considered feeding in different patches).

### Social states across development

If the simulated immature encountered a food item that was not in its diet repertoire, our ABM assigned that immature as being in one of four social states: alone, distant association, close association or peering. The social states of close association and peering relate to possible forms of social learning available to wild immatures (including processes of enhancement and observational social learning, respectively). The probability of a simulated immature being in each social state was determined by a hierarchy of decisions. First, is the simulated immature associated with a conspecific? If not, they are assigned the state alone for that specific feeding patch. Second, if associated with a conspecific, is the simulated immature in close association? If so, for that feeding patch, they are assigned the state close association; otherwise, they are assigned the state distant association. Third, if in close association, is the simulated immature peering? If so, they are assigned the state peering for that feeding patch.

We estimated the probabilities of simulated immatures being in each social state using focal follow data for wild individuals within their first 15 years of life. We also estimated their probability of exploring food items when in each social state across this developmental period (see 'Exploration rates across development').

### Probability of association

To estimate the probability of a simulated immature being associated with a social partner, we counted the number of scans where a focal individual was within 50 m of at least one conspecific (and therefore in association) per follow, versus those in which they were outside this range and classified as alone. We calculated the probability of an immature being associated with a conspecific at different ages using a quasibinomial GLMM (with data points weighted by the total number of scans per follow and with focal ID included as a random intercept). At each age, the probability of being alone was $1 - p_{Age}$(associated).

### Probability of distant or close association

For all scans where an individual was in association, we determined the probability of being in close or distant association. An analysis of our data showed that orangutan exploration rates when within 0–10 m of a conspecific were very similar across ages; however, at greater distances (10–50 m) exploration rates across ages were much lower. We therefore defined close association as being within 10 m of at least one conspecific. For each follow, we divided the number of scans where an individual was associated with a conspecific into those where they were in close and distant association. We then modelled the probability of being in close association when associated with a conspecific at different ages, using a quasibinomial GLMM (with data points weighted by the total number of scans for each follow and with focal ID as a random intercept). At each age, we set the probability of simulated immatures being in close association (if associated with another individual) to be equal to that for wild individuals, and $p_{Age}$(distant association) was $1 - p_{Age}$(close association).

### Probability of peering

When in close association, orangutans may perform close-range observation of the behaviour and/or objects manipulated by conspecifics (peering). To estimate how peering rates change with age, we used a quasibinomial GLMM comparing the number of close association scans where an individual was peering with the number of scans where an individual was not peering (weighted by the total number of scans in close association and with focal ID as a random intercept). We used this rate as the probability of simulated immatures peering when in close association in a feeding patch at a given age.

### Exploration rates across development

The majority of great ape learning is facilitated through exploration, which can be mediated and enhanced through social interactions[27,38–40]. Exploration is defined as repetitive, often destructive manipulation (including failed feeding attempts) of objects (including food items), while the visual and tactile foci of the individual are on the object[27,72]. To quantify the effects of these different social factors on the probability of exploration, we modelled the probability of wild immatures exploring when in distant and close association, as well as after peering. It was not possible to accurately estimate the exploration rates of wild orangutans when alone at all ages, as dependent individuals spend virtually all of their time in association with their mothers. We therefore set simulated immatures' exploration rates when alone to be equal to the exploration rate at distant association for each age.

## Exploration in distant and close association

We estimated how the exploration rates of wild immatures change over development, including when in both distant and close association. For each follow, we sampled all behaviour scans where an immature was in association, and we counted the number of scans where the focal immature was performing an exploration behaviour versus scans describing any other behaviour. We partitioned these counts into those where an individual was in close and distant association. To control for the effect of peering (which can increase exploration rates; see below), we excluded exploration scans where an orangutan had peered at a food item of the same species within the previous hour. We controlled for peering at the level of the species of the food item, because if an immature is drawn to a species of food following peering, they may choose to explore multiple parts (for example, leaves or bark). We modelled the probability of exploration over age for both association categories (close and distant) using a binomial GAMM (with follow number and focal ID as random intercepts and with our model weighted by the total number of scans sampled in each association proximity condition per follow). We translated the model's mean exploration probability at each age into the exploration rates of simulated immatures when in distant and close association.

## Exploration following peering

To estimate exploration rates following peering, we used a subset of the follow data where observers recorded peering events on an all-occurrence basis. For each follow, we recorded each instance (or the last instance in case of peering events in close succession) when an immature peered at a specific species of food item, and whether the immature explored this food item within the following hour (coded as a binary Y/N). We determined whether peering had a significant effect on the exploration rate by comparing the probability of exploring a food item after peering (if the food item had not already been explored in the hour before) with the probability of exploring food items in the hour prior to peering events (a baseline for exploration in contexts that may elicit peering, such as around novel foods). For both of these scenarios, we used binomial GLMMs with focal ID as a random factor. We also estimated whether the effect of peering on exploration probability changed over the course of development. We modelled the proportion of peering events that were followed by exploration behaviours in the subsequent hour, compared with those that were not followed by exploration during the same time frame, for wild immatures across the first 15 years of development (binomial GLMM with focal ID as a random factor and weighted by the number of data points at each age). Model estimates were then translated into the probability of simulated immatures exploring in the peering state at each age.

## Number of explorations needed to learn

The total amount of exposure wild immatures require to learn how to process a food item depends on the number of processing steps that must be performed before it can be ingested[29] (herein considered a food item's complexity). We therefore scaled the number of explorations that simulated immatures require to learn how to consume food items by their complexity.

To estimate the number of times wild immatures explored food items of each complexity category prior to learning how to eat them, we first sampled the five most common food items of each complexity category and determined the earliest age at which wild orangutans were observed consuming each food item using focal follow data (the earliest estimated learning age). We restricted this analysis to immatures who had been followed since at least the age of 1 year, ensuring that the first age that food items were observed being consumed was probably close to the age of learning. To estimate the number of times a species of food item was encountered before an individual learned how to process and eat it, we calculated the rate at which food items occurred in adult focal follows (their probability of being encountered each day). We multiplied this number by the number of days preceding the individual's estimated age of learning (that is, the total number of days spent foraging prior to learning). This provided an estimate of the number of times a particular type of food item was encountered prior to learning. We then estimated the number of times an immature explored a food item prior to learning how successfully eat it, by multiplying the number of encounters by the mean exploration rate across the dependency phase.

Across the five most frequent food items of each complexity category, we averaged the estimated number of explorations required for learning. As more complex food items are rarely encountered, this analysis permitted us to estimate the required number of explorations for complexity categories 0–2 (foods of category 0 required one exploration, category 1 required two explorations and category 2 required two explorations). We extrapolated this trend to higher complexity categories by incremental increases of one exploration for every two increasing steps in complexity (three explorations for category 3, three explorations for category 4 and four explorations for category 5).

## Reporting summary

Further information on research design is available in the Nature Portfolio Reporting Summary linked to this article.

## Data availability

All supporting data are available via Mendeley Data at https://data.mendeley.com/datasets/7kvr22vk5f/3.

## Code availability

All supporting code is available via Mendeley Data at https://data.mendeley.com/datasets/7kvr22vk5f/3.

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

## Acknowledgements

We acknowledge the hard work of all field staff, research assistants and students who have supported long-term data collection at Suaq. We also thank the Indonesian State Ministry for Research and Technology (BRIN-RISTEK); the Ministry of Internal Affairs, Indonesia; the Directorate General of Natural Resources and Ecosystem Conservation–Ministry of Environment and Forestry of Indonesia (KSDAE-KLHK); and the Balai Besar Taman Nasional Gunung Leuser (BBTNGL) in Medan for providing permission to conduct data collection. We thank the Yayasan Ekosistem Letari (YEL) and their the Sumatran Orangutan Conservation Program (SOCP) for hosting our research project at the Suaq Balimbing Monitoring Station. We also thank Universitas Nasional (UNAS) for their collaboration and support. We thank I. Razik for help designing Figs. 2–4. We also thank A. Mielke for his feedback on an initial draft of this paper. Furthermore, we thank R. Young and F. Lamarque for their help curating the Suaq database. Financial support was provided by the Max Planck Institute of Animal Behavior (funding C.S., and in extension, E.H.S.), the SUAQ Foundation (supporting research at Suaq), the University of Zurich (supporting research at Suaq), the A.H. Schultz Foundation (funding awarded to C.S.), the Leakey Foundation (Primate Research Fund and project grant to C.S.) and the Volkswagen Stiftung (Freigeist fellowship to C.S.). All funding for open-access publication was provided by the Max Planck Society. The funders had no role in study design, data collection and analysis, decision to publish or preparation of the manuscript.

## Author contributions

E.H.S. contributed to conceptualization; designed the ABM; conducted all data analysis and programming; produced Figs. 1, 3 and 4, as well as Extended Data Fig. 1 and the Supplementary Tables; and led the writing of this paper. C.T., C.v.S., B.B. and A.W. provided feedback and edited paper. T.M.S. and D.P. contributed to research coordination and commented on the manuscript. C.S. conceptualized the study, acquired funding, provided substantial feedback on and edits to the manuscript, contributed to data collection and offered supervision.

## Funding

## Competing interests

The authors declare no competing interests.

## Additional information

**Extended data** is available for this paper at https://doi.org/10.1038/s41562-025-02350-y.

**Correspondence and requests for materials** should be addressed to Elliot Howard-Spink or Caroline Schuppli.

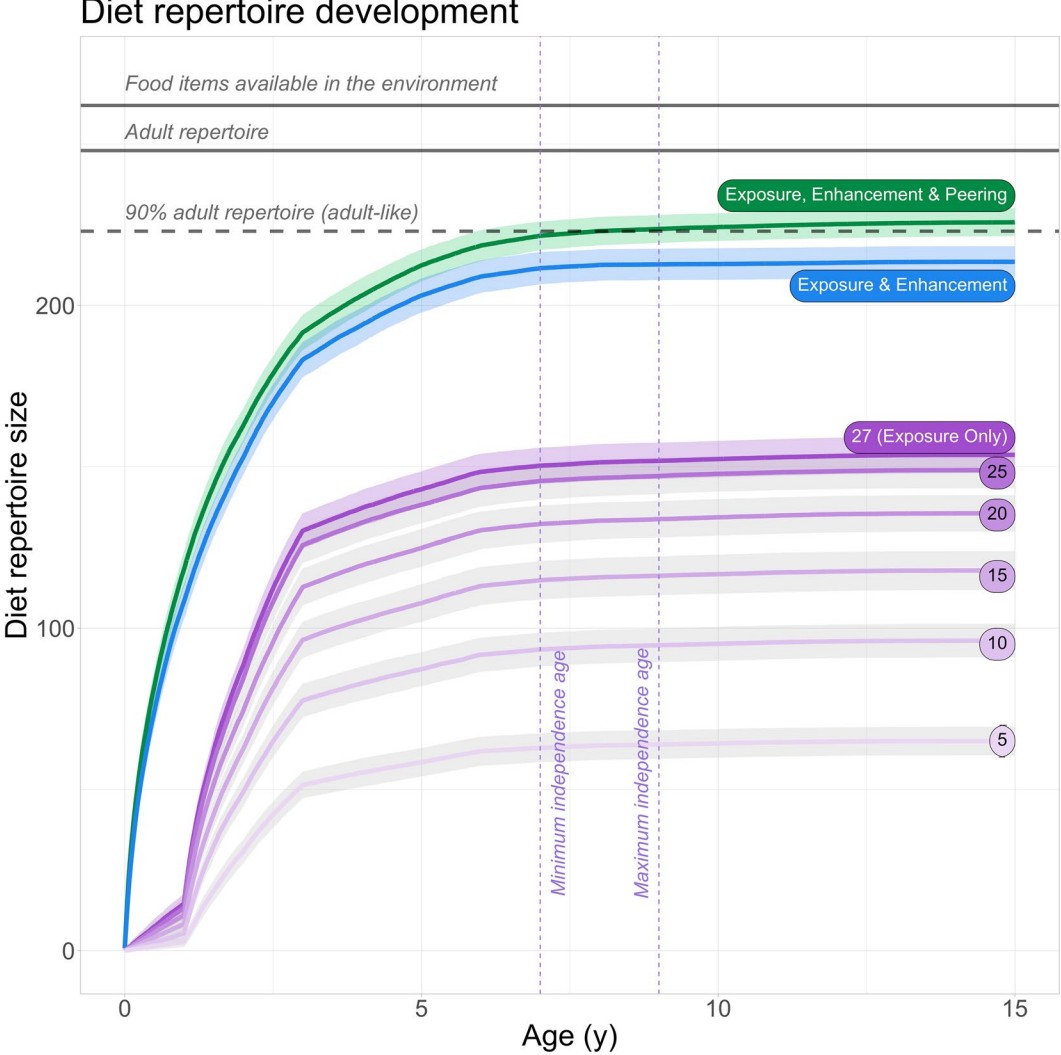

## Diet repertoire development

**Extended Data Fig. 1 | Simulated scenarios within the Zone of No Exposure.** This figure is a recreation of Fig. 4 in the main manuscript, but includes additional scenarios where the number of feeding patches visited by simulated immatures in the 'Exposure Only' treatment was systematically varied (N = 250 simulations for each scenario). The numbers for each simulation depict the mean number of feeding patches simulated immatures visited per day. This graph provides some internal resolution to the 'Zone of No Exposure' (see Fig. 4), by highlighting how reduced opportunities for exposure can also limit diet development; however, it is important to reiterate that these model does not account for individuals' survivability in each scenario. We predict that without maternal influences (thus, reducing exposure), it becomes increasingly unlikely that dependent immatures would survive for long enough to accumulate information about available foods in the wild. Thus, these precise trajectories presented within the 'Zone of No Exposure' should be interpreted with caution.

# Reporting Summary

## Statistics

For all statistical analyses, confirm that the following items are present in the figure legend, table legend, main text, or Methods section.

| n/a | Confirmed | |
|---|---|---|
| ☐ | ☒ | The exact sample size (*n*) for each experimental group/condition, given as a discrete number and unit of measurement |
| ☐ | ☒ | A statement on whether measurements were taken from distinct samples or whether the same sample was measured repeatedly |
| ☐ | ☒ | The statistical test(s) used AND whether they are one- or two-sided *Only common tests should be described solely by name; describe more complex techniques in the Methods section.* |
| ☐ | ☒ | A description of all covariates tested |
| ☐ | ☒ | A description of any assumptions or corrections, such as tests of normality and adjustment for multiple comparisons |
| ☐ | ☒ | A full description of the statistical parameters including central tendency (e.g. means) or other basic estimates (e.g. regression coefficient) AND variation (e.g. standard deviation) or associated estimates of uncertainty (e.g. confidence intervals) |
| ☐ | ☒ | For null hypothesis testing, the test statistic (e.g. *F*, *t*, *r*) with confidence intervals, effect sizes, degrees of freedom and *P* value noted *Give P values as exact values whenever suitable.* |
| ☒ | ☐ | For Bayesian analysis, information on the choice of priors and Markov chain Monte Carlo settings |
| ☐ | ☒ | For hierarchical and complex designs, identification of the appropriate level for tests and full reporting of outcomes |
| ☐ | ☒ | Estimates of effect sizes (e.g. Cohen's *d*, Pearson's *r*), indicating how they were calculated |

*Our web collection on statistics for biologists contains articles on many of the points above.*

## Software and code

Policy information about availability of computer code

| Data collection | No custom code was used for data collection. |
|---|---|
| Data analysis | All data analysis was conducted in R (v 4.5.1 Great Square Root). We used GAMM and GLMM models to analyze data collected from wild individuals. Poisson and binomial GLMMs were constructed using the lme4 package's glmer() function (v. 1.1.37). Quasibinomial GLMMs were fit by Penalized Quasi-Likelihood using the MASS package (v. 7.3.65). GAMMs were fit using the mgcv package (v.1.9.3). Confidence intervals around model estimates were calculated using the parameters package (v. 0.26.0) For binomial models, dispersion tests were run using the testDispersion() function of the DAHRMa package (v 0.4.7). Dispersion of Poisson models were checked using the check_overdispersion() function of the Performance package (v. 0.14.0). A detailed account of the design of all models is provided within the Supplementary Information. Our ABM was programmed using Python (v. 3.8.11), and random numbers were generated using NumPy (v. 1.22.4). All code used for data analysis and to run the ABM can be retrieved using the link under 'Data and Code Availability' within the main manuscript. Versions for all packages and software are included in the supplementary materials. |

For manuscripts utilizing custom algorithms or software that are central to the research but not yet described in published literature, software must be made available to editors and reviewers. We strongly encourage code deposition in a community repository (e.g. GitHub). See the Nature Portfolio guidelines for submitting code & software for further information.

## Data

Policy information about availability of data

All manuscripts must include a data availability statement. This statement should provide the following information, where applicable:

- Accession codes, unique identifiers, or web links for publicly available datasets
- A description of any restrictions on data availability
- For clinical datasets or third party data, please ensure that the statement adheres to our policy

All data supporting our results can be found in the following repository: https://data.mendeley.com/datasets/7kvr22vk5f/3

## Research involving human participants, their data, or biological material

Policy information about studies with human participants or human data. See also policy information about sex, gender (identity/presentation), and sexual orientation and race, ethnicity and racism.

| | |
|---|---|
| Reporting on sex and gender | N/A |
| Reporting on race, ethnicity, or other socially relevant groupings | N/A |
| Population characteristics | N/A |
| Recruitment | N/A |
| Ethics oversight | N/A |

Note that full information on the approval of the study protocol must also be provided in the manuscript.

# Field-specific reporting

Please select the one below that is the best fit for your research. If you are not sure, read the appropriate sections before making your selection.

☐ Life sciences      ☐ Behavioural & social sciences      ☒ Ecological, evolutionary & environmental sciences

For a reference copy of the document with all sections, see nature.com/documents/nr-reporting-summary-flat.pdf

# Ecological, evolutionary & environmental sciences study design

All studies must disclose on these points even when the disclosure is negative.

| | |
|---|---|
| Study description | We leveraged long term data collected at the Suaq Balimbing research area, Sumatra Indonesia. Data was collected on the behaviors of wild Sumatran orangutans (Pongo abelii). All individuals included in this study are well known, and form part of long-term data collection at Suaq. |
| Research sample | Our dataset on wild orangutans included data collected from 2676 focal follows on 132 individuals, totaling 22,547 hours of observation time. Each analysis within our manuscript used a different subset of this data based on the data's relevance. For example, when estimating adult diet repertoire size we only used data from individuals who had reached sexual maturity (Adults = 95; Follows = 1620; Scans = 402,082). We provide a full list of sample sizes for each analysis (including at different levels of the data where relevant, such as the number of individuals from whom we collected data for a given analysis, and the total number of behavioral scans sampled across these individuals). |
| Sampling strategy | Within the forest, focal orangutans were followed opportunistically upon encounter. Once encountered, orangutans continued to be followed throughout the day until they made night nests. Orangutans were followed for a maximum of 10 consecutive days, after which another individual was saught out for sampling. Our data is therefore both long-term and cross-sectional, as not all individuals can be followed on each day of data collection. We sampled and analyzed all data collected between 2007-2019. |
| Data collection | Data was recorded on paper or digital tablets. Data was recorded on focal orangutan behaviors at 2-minute intervals by a trained team of researchers and field assistants via instantaneous sampling. Instantaneous sampling recorded all behaviors performed by orangutans, as well as any objects or food items associated with these behaviors. These behaviors included all instances of exploration (including solo object play, and failed feeding attempts) as well as feeding and peering behaviors. Additional data was collected on peering behaviors in wild orangutans on an all-occurence basis by one observer (C.S.) who specifically collected data on how peering behaviors related to exploration of target food items in the hour before and after peering. In both sampling strategies, the target of orangutans' observations during peering (including the individual being peered at, the target individual's behavior, and any objects they are manipulating) were recorded recorded. A link to the full protocol for data collection at Suaq is provided as part of our online methods. |

| | |
|---|---|
| Timing and spatial scale | We sampled and analyzed all data collected between 2007-2019. All data was recorded at Suaq Balimbing Monitoring Station in South Aceh, Sumatra, Indonesia, which covers 550ha, at around 5m above sea level. |
| Data exclusions | We only included data from observers who passed minimum thresholds of inter-observer reliability. For each analysis, we only used data which was relevant to the question at hand (see example above on adult diet-repertoire size). Any data which was filtered for each analysis is clearly described within the methods and results of our manuscript, including sample sizes after filtering. |
| Reproducibility | N/A, data were collected through day-long observations of wild animals. |
| Randomization | N/A, data were collected through day-long observations of wild animals, without interacting with the animals, or assigning them to any experimental 'treatments'. |
| Blinding | N/A, the animals were individually known to the observers, and this was a fundamental aspect of long-term data collection at the individual level at Suaq. |

Did the study involve field work?  ☒ Yes  ☐ No

## Field work, collection and transport

| | |
|---|---|
| Field conditions | Data was collected at the Suaq Balimbing Monitoring Station in South Aceh, Sumatra, Indonesia. This research area covers a deep peat swamp forest. There is standing water covering the area year round, which raises during wetter seasons (two distinct periods per year). At the centre of the research area, water can reach chest height. Temperatures range from 19-38 degrees celsius each day. |
| Location | Data was collected at the Suaq Balimbing Monitoring Station in South Aceh, Sumatra, Indonesia (3° 02.873' N, 97° 25.013'E). This research area covers 550ha and is approximately 5m above sea level. |
| Access & import/export | This field site can be accessed by boat. via the Lembang River. No physical samples were collected during this study. |
| Disturbance | No disturbance was experienced during the study period. All observers take specific care not to disturb orangutans during observation. Given that orangutans are arboreal, observers always maintain a suitable distance from observed individuals. |

# Reporting for specific materials, systems and methods

We require information from authors about some types of materials, experimental systems and methods used in many studies. Here, indicate whether each material, system or method listed is relevant to your study. If you are not sure if a list item applies to your research, read the appropriate section before selecting a response.

## Materials & experimental systems

| n/a | Involved in the study |
|---|---|
| ☒ | ☐ Antibodies |
| ☒ | ☐ Eukaryotic cell lines |
| ☒ | ☐ Palaeontology and archaeology |
| ☐ | ☒ Animals and other organisms |
| ☒ | ☐ Clinical data |
| ☒ | ☐ Dual use research of concern |
| ☒ | ☐ Plants |

## Methods

| n/a | Involved in the study |
|---|---|
| ☒ | ☐ ChIP-seq |
| ☒ | ☐ Flow cytometry |
| ☒ | ☐ MRI-based neuroimaging |

## Animals and other research organisms

Policy information about studies involving animals; ARRIVE guidelines recommended for reporting animal research, and Sex and Gender in Research

| | |
|---|---|
| Laboratory animals | No laboratory animals were used in our study. |
| Wild animals | We collected observational data on a population of wild Sumatran orangutans (Pongo abelii) which are habituated to human presence since the inception of research in 2007. No animals were captured or transported as part of this study. All data was collected non-invasively through observation. The age of animals observed varied from less than 1 year old to (what we estimate to be) over 60 years of age, and long term data collection includes the collection of data on both male and female animals. These animals continue to live freely in the wild. |
| Reporting on sex | N/A |
| Field-collected samples | No sampled materials were taken from this field site. |

| Ethics oversight | The Indonesian State Ministry for Research and Technology (RISTEKDIKTI) and the National Research and Innovation Agency (BRIN). See our Ethics Statement. |

Note that full information on the approval of the study protocol must also be provided in the manuscript.

## Plants

| Seed stocks | No seed stocks were used in this study. |

| Novel plant genotypes | No novel plant genotypes were generated in this study. |

| Authentication | N/A |

