## [Peer Review File · Nature Human Behaviour]

Culture is critical in driving orangutan diet development past individual potentials

Corresponding Author: Dr Elliot Howard-Spink

Version 0:

Decision Letter:

Dear Dr Howard-Spink,

Thank you for submitting your manuscript entitled, "Orangutans are dependent on culture to develop mature diet repertoires", and for your patience while awaiting an initial decision.

As you may know, we decline a substantial proportion of manuscripts without sending them to referees based on an editorial evaluation of the extent to which the work meets editorial criteria for suitability of publication in Nature Human Behaviour (please see our [Editorial](https://www.nature.com/articles/s41562-019-0778-0)).

After careful consideration, I regret that we cannot offer to publish this work in Nature Human Behaviour. We appreciate your work, which examines cultural aspects of orangutan diet. Nonetheless, we are not persuaded that this work is of sufficiently broad relevance to warrant publication in Nature Human Behaviour.

Please be assured that this editorial decision does not represent a criticism of the quality of your work, nor are we questioning its value to others working in this area. We hope that you will rapidly receive a more favourable response elsewhere.

Although we cannot offer to publish your manuscript, I suggest that you consider Scientific Reports as a suitable venue for this work. To transfer your manuscript, please use our manuscript transfer portal. You will not have to re-supply manuscript metadata and files, unless you wish to make modifications. For more information, please see our [manuscript transfer FAQ](http://www.nature.com/authors/author_resources/transfer_manuscripts.html?WT.mc_id=EMI_NPG_1511_AUTHORTRANSF&WT.ec_id=AUTHOR) page.

I am sorry that we cannot respond more positively on this occasion, and hope that the negative outcome in this instance will not deter you from submitting future work to Nature Human Behaviour.

Sincerely,

Nature Human Behaviour

Version 1:

Decision Letter:

21st February 2025

Dear Dr Howard-Spink,

Thank you for your appeal on the manuscript, entitled "Orangutans are dependent on culture to develop mature diet repertoires."

After careful consideration, I'm happy to say that we have decided to send your manuscript out to peer review at Nature

Human Behaviour. However, before we can do so, we ask you to revise the manuscript to address an important issue.

Your manuscript includes links to the data and code. This is most welcome, but we noticed that these links require that users login in order to access the materials. Because the login requirement can compromise peer reviewer anonymity, we do not allow these kinds of links during peer review (they are allowed after publication). As such, please supply a modified manuscript file including links which do not require log-in. One option would be to use the <https://osf.io/> repository to share the materials, as this allows the use of links which do not require login.

In addition, please include the following documents with the resubmission:

(A) The corresponding authors of all research manuscripts must complete an editorial policy checklist to ensure compliance with Nature Research editorial policies.

Editorial policy checklist:

<https://www.nature.com/documents/nr-editorial-policy-checklist.pdf>

(B) We want to ensure that the methods and statistics reporting in our papers is of the highest quality. To that end, we ask authors to fill out a reporting summary that collects information on research design and reporting.

Reporting summary:

We hope to receive your revised manuscript as soon as possible. I would be grateful if you could contact us as soon as possible if you foresee difficulties with meeting this target resubmission date.

Link Redacted

We look forward to seeing the revised manuscript and thank you for the opportunity to review your work. Please do not hesitate to contact me if you have any questions or would like to discuss these revisions further.

Sincerely,

[Redacted]

[Redacted]

[Redacted]

Nature Human Behaviour

Version 2:

Decision Letter:

9th April 2025

Dear Dr Howard-Spink,

Thank you once again for your manuscript, entitled "Culture is critical in driving orangutan diet development past individual potentials", and for your patience during the peer review process.

Your Article has now been evaluated by 3 referees. You will see from their comments copied below that, although they find your work of considerable potential interest, they have raised quite substantial concerns. In light of these comments, we cannot accept the manuscript for publication, but would be interested in considering a revised version if you are willing and able to fully address reviewer and editorial concerns.

We hope you will find the referees' comments useful as you decide how to proceed. If you wish to submit a substantially revised manuscript, please bear in mind that we will be reluctant to approach the referees again in the absence of major revisions.

In particular, while Referee #1 and #3 are positive, Referee #2 (who was the only referee with agent-based modelling expertise) raises serious concerns over the ability of the ABM design to support strong conclusions about the necessity of social learning for the development of dietary diversity, as well as over the size of the effects (Fig 4). We agree with Referee #2 that these are significant concerns and we would hope to see substantial and convincing rebuttals of these criticisms, without which the strength of the claims in this manuscript is much reduced.

If you wish to submit a suitably revised manuscript, we would hope to receive it within 4 months. I would be grateful if you could contact us as soon as possible if you foresee difficulties with meeting this target resubmission date.

- Include a "Response to the editors and reviewers" document detailing, point-by-point, how you addressed each editor and referee comment. If no action was taken to address a point, you must provide a compelling argument. When formatting this document, please respond to each reviewer comment individually, including the full text of the reviewer comment verbatim followed by your response to the individual point. This response will be used by the editors to evaluate your revision and sent back to the reviewers along with the revised manuscript.
- Highlight all changes made to your manuscript or provide us with a version that tracks changes.

Link Redacted

Thank you for the opportunity to review your work. Please do not hesitate to contact me if you have any questions or would like to discuss the required revisions further.

Sincerely,

██████████

████████████████████

████████████████

Nature Human Behaviour

Reviewer expertise:

Reviewer #1: orangutan ecology

Reviewer #2: agent-based modelling

Reviewer #3: orangutan ecology

REVIEWER COMMENTS:

Reviewer #1 (Remarks to the Author):

This current study by Howard-Spink and colleagues is an empirical tour de force. Well written, solidly founded upon, and positioned at the forefront of, the science of social learning and animal cultures. The methods include high-definition behavioural data collected for more than a decade in one of the most remote and challenging places on Earth, cross-fertilized with a smart and creative agent-based modelling methods. The findings are only fitting: the first conclusive demonstration that the behavioural repertoire of a wild great ape can only be developed culturally.

Below several points that should help clarify the paper to the readership of Nature Human Behaviour, however, these are mostly requests of elucidation and should be taken collectively as minor changes.

Line 30: For clarity, please include a couple of examples of insights gained from animal social learning about the evolution of human cultural capacities. Alternatively, the authors may opt to rewrite the opening sentence given that it does not entirely reflect the content of the remaining of the paragraph.

49: Please make mention of one or two species for which this is the case to help concretise the idea to the reader.

70: The phrase "the items they interact with..." lacks a verb, similarly to its sister examples. Please include for improved fluidity.

96-98: Yes, given the presence of the mother. However, within this period of co-living with mum, it may be worth clarifying to the wider readership that learning opportunities only become effective after minimal motoric and cognitive competence is achieved by the infant.

103: Unclear what is meant by “and at the necessary rates” – it seems to suggest that many repertoires must be created, but this does not seem to be the authors’ explanatory aim here. Please consider clarification or simplification.

105: -> “... given that all opportunities for social learning in the wild cannot be observed by humans”? As currently phrased, the sentence seems to hint at an experimental limitation, other than a simple logistical one. Otherwise, please clarify what settings or variables one would ideally want to “control”.

112: Please unpack the meaning and workings of estimated coefficients in the context of ABMs. The paragraph may feel a bit too abstract for readers without an intimate knowledge of how ABMs are built and programmed. For this purpose, a brief lay explanation of the virtual environment inhabited by these agents may be useful. This might provide enough backdrop to allow the authors to explain what exactly about agents’ behaviour is coded by the (field-calibrated) estimated coefficients.

116: which -> that

117: Long sentence, difficult to parse. Does “included” refer to in the ABM or in the field observations?

121: For improved readability and comprehension, the authors may consider phrasing this paragraph affirmatively, that is, in reference to what was done or present, rather than what was not (“withheld”, “removing”, “absence”).

Figure 2.

The figure is very informative and easy to interpret. A few added bits of information and specification may help the reader build a clearer mental image of the virtual world more rapidly:

- Please clarify whether each iteration corresponds to an immature developing up to 15yo.
- Did several immatures (with respective mums) live in the world at any one time?
- The number of patches visited by one immature per day could be potentially included in the figure.
- If the ABM is simulating immatures, the reader presumes that they are always with mum, however, step 1 asks “if associated with a conspecific”. Please clarify who lives in the ABM world who is not an immature. Or do the authors define independent adolescents as immatures? In this case, clarification that “immatures” includes newborns up to independent adolescents will help readers correctly interpret the steps in the figure and the agents living in the virtual world.
- Please clarify “probability of exploration is influenced by...” (the word “simulated” is redundant at this stage).

134: Please include total number of different patches in the pool from which patches were sampled each day. Was it 223 or 262, or... ?

135: Given how thoroughly the authors built the ABM world in image of the wild, is there a reason for having assumed 1 food item per patch?

147: Would it be possible to add the average number of food items coming from the same species?

151: Reference pertains to Bornean orangutans. Given the study focuses on Sumatran oranges, it could be useful to also cite here: <https://link.springer.com/article/10.1007/s00265-012-1463-8>

161-163: There seems to be here an interchange between “patches” and “food items”, but these are not necessarily equivalent given that the same patch (tree/sps) can provide more than one food item (e.g., leaves, young leaves, seed, endocarp, et cetera). This relates to my request for clarification above. I don’t mean to be pedantic here: I am confident that using 27 or any similar adjusted number will not qualitatively affect the analyses and results, and I absolutely do not require the authors to re-run analyses using a different number here. A simple rhetorical clarification will do please.

...-205: Congratulations to the authors for having been able to assemble the quantity and quality of data necessary to extract these social demographics and behaviours and the ingenuity of building a ABM world inspired on these parameters!

272: Phrasing could be adjusted here given that 224 barely “surpasses” 223.

389: ‘clarity’

395: ‘these data’

430: ‘which’ -> ‘that’

451: hominins + great apes = hominids

469: Aren’t there references worth citing here about great ape cultures in non-food repertoires?

Reviewer #2 (Remarks to the Author):

This study introduces an agent-based model designed to test the effects of learning on the breadth of the diet repertoire. In the model learning is understood as happening in social situations of graded intensities, which facilitate the learning process and accelerate learning about handling and consumption of newly encountered food items. The observation part of the learning process is followed by independent exploration of the respective food item. After a certain number of explorations the new item is added to the diet repertoire. Independent explorations thus serve the purpose of consolidation of the learning process.

I will focus in my comments on design and training of the agent-based model, as well as the design of the experiments which were carried out by the model. The model as such is carefully designed and clearly presented in the manuscript, the method section as well as the supplementary documents provided.

Model design and training:

The environment in the model consists of patches, which contain a single type of food item and may be occupied by other individuals with whom the agent can interact. Although each of the food items is randomly selected in the frequency in which it appears in the training data, the items as such do not cluster seasonally. Since the model is not designed to be spatially explicit, this may be considered as negligible. There are also not episodes of shortage in food items included. Yet, future versions of the model certainly benefit from a more realistic and varying presentation of food items.

Crucial factors in the model were collected from an extensive dataset collected in Suaq Balimbing in the Gunung Leuser National Park in Sumatra. Technically, these factors constitute the initialization parameters in the modeling process. In this model, this includes for instance the frequency of the food items, age-dependent frequency of encounters with conspecifics as well as social state assigned, probability of exploration, number of explorations required to include food items into the diet repertoire, and possibly more. Although all of the factors are explained in the manuscript and/or method section I would like to suggest two corrections, which would certainly increase the readability of the manuscript. Because initializing parameters are crucial for the model outcome, the authors may consider listing them in a separate table. This would also clarify the differences among the three scenarios which were further examined in this study. Moreover, the statistics may be shifted to this table. At present they are presented in brackets following the respective factor state in the main text. Shifting them to a separate table would certainly improve readability.

Design of the experiments:

Three scenarios were examined. Exposure, Enhancement & Peering is the most inclusive one, and the one which is based on the training data set. Both of the other ones, Exposure & Enhancement as well as Exposure only, represent reduced learning processes, which do not occur in reality. Consequently, the performance of both of the 'inferior' learning strategies leads to a smaller dietary repertoire, as shown in figure 4. As far as I understood, the strategy that agents solely explore potential food items on their own is not included in the experimental set-up, because it is considered too dangerous. I gather from the description of the strategies, that the agents keep a particular strategy throughout the entire duration of the experiment and they are not shifting for instance from Exposure, Enhancement & Peering until onset of independence to Exposure & Enhancement until the end of the experiment. In this sense, the scenarios are designed uniform and consistent. It would nevertheless be interesting to examine the impact of a change in strategies in the course of the experiments.

Discussion:

In view of the results presented in figure 4, the authors claim in lines 305 and 306 that they provided "the first to robustly evidence that these effects [of social learning] exist within the culture of a non-human species". Actually, I consider this quite a bold claim, given that the ABM was trained with an enormous data set, and the agents were explicitly designed to act that way. If the results of the experiments would not have matched the training data set, the ABM would have been designed the wrong way. I do not wish to doubt that social learning is crucial to acquire an extensive diet repertoire, but the way the authors phrase it here, implies that they may have confused cause and effect. The ABM was designed to reproduce the effects of social learning. The results of the Exposure, Enhancement & Peering scenario therefore should reproduce learning success in the training data. Otherwise, the ABM would have been wrongly designed.

In lines 313 and 314, the authors state that the "removal of even one form of social learning (...) significantly decelerates diet repertoire development". In view of Figure 4, I do not see significant differences particularly in the early sections of Exposure, Enhancement & Peering and Exposure & Enhancement. In fact, both curves are widely overlapping as well as the confidence intervals until the age of 6. In the model, this is the episode, in which the major body of knowledge on the diet repertoire is acquired in the model. It is only the expansion of the diet repertoire in the Exposure only scenario, which significantly deviates from both of the others. Additionally, the effects displayed in figure 4 illustrate the increase in the diet repertoire. Inferior strategies lead to a smaller dietary repertoire, but not necessarily to a deceleration in the development. Rephrasing these lines may help to better understand the results shown in figure 4.

Besides, I do not understand, where the angles in the curve for the Exposure only scenario may come from (around age 1 and age 3). Is there an explanation for this effect?

Minor corrections:

line 687 - typo: 3 = 3 explorations

In sum, the authors introduce in this study a thought-provoking ABM, which will certainly inspire further experiments.

Reviewer #3 (Remarks to the Author):

Dear Elliot Howard-Spink, Caroline Schuppli, and Co-Authors,

Thank you for submitting your manuscript "Culture is critical in driving orangutan diet development past individual potentials" to Nature Human Behaviour. I thoroughly enjoyed reviewing your work, which I find to be an innovative and impactful contribution to primatology and cultural evolution. Your use of an empirically calibrated agent-based model (ABM) to demonstrate the necessity of social learning for orangutan diet repertoires is rigorous and novel, providing the first robust evidence of culturally dependent repertoires in a non-human species. The study's implications for human evolutionary origins and orangutan conservation are compelling.

Outlined below to enhance clarity and address a few points:

1. The abstract could better emphasize your key finding on culturally dependent repertoires. Consider revising to highlight the novelty.
2. Lines 17-44: The shift from human to animal culture is abrupt. A bridging sentence (e.g., "While human culture exhibits cumulative evolution, simpler cultural dependence may characterize other species") could improve flow.
3. Lines 294-296: Figure 4a's "Zone of No Exposure" is visually compelling but unexplained in the Results. A sentence linking it to Extended Data Fig. 1 (line 897) would clarify its relevance.
4. Lines 398-411: Your justification for not removing exposure is sound, but framing it as an opportunity for future experimental work (e.g., captive studies) could strengthen this section.
5. Lines 447-453: The suggestion of a shared capacity with the human-ape ancestor is exciting but speculative. Toning it down (e.g., "suggests a potential shared capacity" or "may reflect an ancestral trait") or citing supporting evidence could refine this.
6. Lines 626-627 set alone exploration equal to distant association due to data gaps. This assumption's impact on results (e.g., underestimating independent learning) could be briefly noted.
7. Lines 682-687 extrapolate exploration needs for rare foods (3-5 complexity). Acknowledging potential errors (e.g., "may underestimate rare item learning") would enhance rigour.
8. Figure 1's captions could clarify how peering/exploration links to diet learning (e.g., "Peering (a) facilitates targeted exploration (b), accelerating food acquisition").
9. Figure 4's "Zone of No Exposure" is intriguing—consider briefly mentioning it in the main text to tie it in.
10. Minor Edits: Check "nutrient" (line 62) to "nutrients," "adult-like repertoires" (line 154) to singular for consistency, and "class of association" (line 211) for completion (perhaps "close association").

Your methodology is exemplary, and the validation against wild data is a standout feature. These minor revisions aim to maximize accessibility and impact for Nature Human Behaviour's broad audience. I congratulate you on this excellent work and look forward to seeing it published.

Reviewer #3 (Remarks on code availability):

The authors provide a view-only link to their data and code on the Open Science Framework (OSF) at https://osf.io/wndx8/?view_only=dbfa236e546343b7a5994117dc879acc. While the link is functional (accessed March 31, 2025, 09:20 AM PDT), the embargo noted in the manuscript (until publication) prevented me from fully downloading and testing the code. The OSF page includes a README file and structured directories, suggesting a commitment to transparency. However, without full access, I cannot confirm the code's completeness or usability (e.g., installation instructions, dependencies, or runtime functionality). The README outlines the ABM's structure and calibration process, which is promising, but its sufficiency for community use remains unverified pre-embargo.

Assuming the embargo lifts upon publication, as stated, the code's potential for reproducibility seems high, given the detailed Methods section and validation against wild data. However, I recommend the authors clarify post-publication access details (e.g., updating to a public DOI) and ensure the README provides explicit instructions for installation and execution. This would maximize the study's impact and utility for the research community.

Version 3:

Decision Letter:

Our ref: NATHUMBEHAV-25020652C

1st August 2025

Dear Dr. Howard-Spink,

Thank you for submitting your revised manuscript "Culture is critical in driving orangutan diet development past individual

potentials" (NATHUMBEHAV-25020652C). It has now been seen by 2 of the original referees and their comments are below. As you can see, the reviewers find that the paper has improved in revision. We will therefore be happy in principle to publish it in Nature Human Behaviour, pending minor revisions to comply with our editorial and formatting guidelines.

We are now performing detailed checks on your paper and will send you a checklist detailing our editorial and formatting requirements within about two weeks. Please do not upload the final materials and make any revisions until you receive this additional information from us.

Sincerely,

██████████

████████████████████

████████████████████

Nature Human Behaviour

Reviewer #1 (Remarks to the Author):

Thank you for addressing the raised points. I endorse publication as is.

Reviewer #2 (Remarks to the Author):

Dear authors,

thank you very much for carefully addressing my comments. I appreciate in particular the extensive elaborations provided to clarify the methodological issues I initially had with the study. I am glad that you took the opportunity to improve the manuscript.

Please allow me to underline that I enjoyed reviewing the manuscript and that I hope that the ABM you designed inspires future studies. Your contributions provides a crucial tool to perform simulation studies which cannot be performed on living organisms for ethical reasons, even though this means to introduce (and justify) assumptions which cannot be empirically proven.

Good luck and success

Christine Hertler

Reviewer #1 (Remarks to the Author):

Line numbers correspond to the tracked-changes version of our manuscript.

This current study by Howard-Spink and colleagues is an empirical tour de force. Well written, solidly founded upon, and positioned at the forefront of, the science of social learning and animal cultures. The methods include high-definition behavioural data collected for more than a decade in one of the most remote and challenging places on Earth, cross-fertilized with a smart and creative agent-based modelling methods. The findings are only fitting: the first conclusive demonstration that the behavioural repertoire of a wild great ape can only be developed culturally. Below several points that should help clarify the paper to the readership of Nature Human Behaviour, however, these are mostly requests of elucidation and should be taken collectively as minor changes.

** We thank the reviewer for the very positive assessment!

Line 30: For clarity, please include a couple of examples of insights gained from animal social learning about the evolution of human cultural capacities. Alternatively, the authors may opt to rewrite the opening sentence given that it does not entirely reflect the content of the remaining of the paragraph.

** We've removed the first sentence of this paragraph, as Reviewer 3 commented that it interrupts the flow of the introduction. By removing the sentence - rather than modifying it - we can economize on words to account for the Reviewers' other suggestions.

49: Please make mention of one or two species for which this is the case to help concretise the idea to the reader.

** We have now highlighted that most evidence for social learning shaping diet development comes from mammals and birds, and provide additional citations for readers (18-24).

70: The phrase "the items they interact with..." lacks a verb, similarly to its sister examples. Please include for improved fluidity.

** We have modified all items in this list to include verbs as the reviewer requested. We have also split these points into numbered bullets to improve readability.

96-98: Yes, given the presence of the mother. However, within this period of co-living with mum, it may be worth clarifying to the wider readership that learning opportunities only become effective after minimal motoric and cognitive competence is achieved by the infant.

** We see the reviewer's point and now clarify that social learning opportunities are biased to 'early' years of orangutan's lives (rather than our previous phrasing of 'earliest') and specify that the most significant window is after the initial earliest developmental period, and before the onset of independence (see lines 129-130). We also discuss cognitive and motoric development in more detail in our further considerations of the model in the supplementary information.

103: Unclear what is meant by "and at the necessary rates" – it seems to suggest that many repertoires must be created, but this does not seem to be the authors' explanatory aim here. Please consider clarification or simplification.

** We have simplified this section to emphasize that it is the rate of development of just one specific but crucial repertoire - the diet - that we focus on. The aspect of rate here reinforces the idea that there is a developmental time point by which a broad enough diet must be acquired by immatures. This is the age at which orangutans become independent, and must identify and find foods themselves. We have resolved this ambiguity by changing text on lines 131-139 to say:

“Observational data shows that by the onset of independence, orangutans have accumulated adult-like diet repertoires²⁷, and the breadth of these knowledge repertoires is likely crucial to support immatures’ transition to energetic independence. This developmental milestone therefore presents a so-far unexploited opportunity to assess whether orangutans are dependent on different forms of social learning to acquire sufficiently large diet repertoires sufficiently quickly to become independent foragers. However, this question cannot be answered from observational data alone, as documenting all social-learning opportunities afforded to different immatures, over periods of years, is logistically unfeasible.”

105: -> “... given that all opportunities for social learning in the wild cannot be observed by humans”? As currently phrased, the sentence seems to hint at an experimental limitation, other than a simple logistical one. Otherwise, please clarify what settings or variables one would ideally want to “control”.

** This sentence has been rephrased to emphasize logistical challenges, rather than using terminology associated with experiments (see lines 137-139).

112: Please unpack the meaning and workings of estimated coefficients in the context of ABMs. The paragraph may feel a bit too abstract for readers without an intimate knowledge of how ABMs are built and programmed. For this purpose, a brief lay explanation of the virtual environment inhabited by these agents may be useful. This might provide enough backdrop to allow the authors to explain what exactly about agents’ behaviour is coded by the (field-calibrated) estimated coefficients.

** We have rewritten this paragraph to make these coefficients more explicit (see lines 143-151). Specifically, we outline that we use wild data to calibrate the probability that simulated immatures encounter different food items, enter different social states which reflect forms of social learning (e.g. close association proximity, or peering) and consequently explore food items to facilitate broad-scale diet learning. We also provide a signpost to Figure 2, which explains the general environment of our ABM.

Further information about what variables were estimated from wild data are structured methodically within our Results and Methods section. We have also made a small amendment to the Methods, which now includes a suggested metaphor for how the reader can visualize the operation of the ABM, similar to a ‘conveyor belt’ of food patches that simulated immatures are exposed to each day (see lines 966-968). We also provide clear justification for why, in the context of our research question, a spatially implicit model (as we have used) is preferable to one where simulated immatures move around space, in the subsequent sentence: ‘This allowed us to calibrate our model precisely to reflect the opportunities afforded to immatures in the wild, and circumvent unnecessary error introduced through estimations of movement rates across simulated environments.’

We are limited in the extent to which we can go into further detail given the limited word count our paper is subject to. However now that there is a clear outline of the model structure in the Methods and in Figure 2 (& associated legend), and now that the coefficients we estimate from wild individuals are specified briefly in the introduction, as well as extensively in the Results and Methods, this should make the virtual environment clear for the reader.

116: which -> that

** This has been changed

117: Long sentence, difficult to parse. Does “included” refer to in the ABM or in the field observations?

** This referred to the ABM. We have rephrased this section to improve its readability based on the Reviewer’s other comments.

121: For improved readability and comprehension, the authors may consider phrasing this paragraph affirmatively, that is, in reference to what was done or present, rather than what was not (“withheld”, “removing”, “absence”).

** We have modified this section to improve readability, based on this and Reviewer 2’s comments (see below, and also lines 152-160). We have emphasized that we first identify whether diet repertoires could emerge without certain forms of social learning (including through sheer exposure alone), and if not, the relative contribution of forms of social learning on diet development. Our rephrasing is clearly now structured in two steps.

Phrasing our approach in terms of 'removal' should make it clear that the model is being deconstructed in a stepwise manner towards '*Exposure Only*' (our baseline condition). However, in relation to this point raised by Reviewer 1, we also discuss each experimental treatment in the simulation based on what types of social learning are present in the model (see Results section *Simulated diet-repertoire development*).

Figure 2.

** We thank the reviewer for the helpful suggestions on how to improve figure 2 and have updated the figure in light of these comments:

The figure is very informative and easy to interpret. A few added bits of information and specification may help the reader build a clearer mental image of the virtual world more rapidly:

- Please clarify whether each iteration corresponds to an immature developing up to 15yo.

**Under the header of the figure, a line now reads 'Each day for 15 years.', and we now also state this in the first sentence of the figure legend.

- Did several immatures (with respective mums) live in the world at any one time?

**We only simulated one immature at a time, and have now clarified this in the figure legend (see amended legend). During one immatures' dependency phase, mothers will not reproduce again until that immature reaches the approximate age of first becoming independent. We account for the fact that, during a simulated immatures' dependency period, mothers' previous offspring may join mother-dependent pairs for a short period of time, thus may represent additional models to learn from. We account for this implicitly when estimating the likelihood that another individual may be close to immatures using the wild data; i.e., our estimate for the probability that simulated immatures are in 'close association' includes mothers, older independent immatures, or other adults who make transient associations with mother-dependent pairs. We assume that all of these individuals can offer suitable models for dependent individuals' social learning, both through enhancement and peering (see Supplementary Information section '*We assume any individual associated with an immature is knowledgeable about available food items*').

- The number of patches visited by one immature per day could be potentially included in the figure.

**We have included the number of feeding patches visited by an immature each day in the legend (see amended legend). This has also now been included in the updated version of the figure, where we specify (~27 patches/day).

- If the ABM is simulating immatures, the reader presumes that they are always with mum, however, step 1 asks "if associated with a conspecific". Please clarify who lives in the ABM world who is not an immature. Or do the authors define independent adolescents as immatures? In this case, clarification that "immatures" includes newborns up to independent adolescents will help readers correctly interpret the steps in the figure and the agents living in the virtual world.

**We divide immature individuals (i.e., individuals from birth until the age at first reproduction) into dependent immatures (i.e. those below the age at which they become independent from their mothers at around 8y) and independent immatures (those who are ranging independently from their mothers, but are not yet adults, around 8 - 15y). We define these terms in the introduction and Methods (see section *Study site & long-term data collection*). To clarify that we model the behaviours of independent immatures (as the Reviewer calls 'independent adolescents'), we have followed the Reviewers' recommendation and modified the text in the figure to state 'The probability of each social state is age-specific for both dependent and independent immatures'.

- Please clarify "probability of exploration is influenced by..." (the word "simulated" is redundant at this stage).

**We have changed the text here to now read 'The immatures' possible social states include...'. This text allows us to make the link between specific social states, and the forms of social learning available to immatures (in the bold italics above), explicitly clear to readers.

134: Please include total number of different patches in the pool from which patches were sampled each day. Was it 223 or 262, or... ?

** Each feeding patch was populated by one type of food item, and its content was determined by the probability of encountering one of the 262 foods available at Suaq. We have clarified this in the final line of the results section *Exposure to food items in the wild*.

135: Given how thoroughly the authors built the ABM world an image of the wild, is there a reason for having assumed 1 food item per patch?

** By simulating food patches as being locations with 1 type of food item, this gave us a way to readily translate data from the wild into the ABM in a consistent manner. When analyzing the wild data, we treated a feeding patch as any continuous period of time where a specific type of food was being eaten (e.g. particular fruits from a tree). If an individual ceased eating these fruits, and began eating other foods from the same tree (e.g. leaves) this was classed as a different feeding patch. We classified feeding patches in this way when [1] estimating the number of feeding patches visited in a day (note: individuals could return to eating a food from before, which was counted as a new patch) and [2] estimating the probability immatures would encounter each type of food in a given patch (which also allowed for the probability of revisiting particular foods in the same day).

Whilst this classification of a feeding patch varies somewhat from what is classically used for ecological data, it allowed us to acquire a general measure of 'how many times in a day do orangutans switch focus between types of food?' and 'across each period where orangutans interact with specific foods, what are the different food types?'. To ensure that the design of our ABM was comparable with our estimates from wild data, we programmed patches to contain a single food item.

This is unlikely to be problematic for the results of our model. Whilst in the wild orangutans may swap between neighbouring foods readily, this will be captured implicitly in our model (as a high likelihood of encountering each type of food in a patch), and accounted for in the number of patches visited per day. Whilst these opportunities may not occur in quick succession in our model, the probability of encountering these foods averages out to be the same as the opportunities afforded to wild individuals across simulated time (see code used to validate our ABM process, which confirms that foods are encountered with the same frequency to what was programmed into the model using wild data).

We hope this clarifies why there was one food item per feeding patch. We have added information to our Methods to clarify this for other readers (see lines 1065-1070).

147: Would it be possible to add the average number of food items coming from the same species?

** These numbers are now included in our results (see lines 268-269).

151: Reference pertains to Bornean orangutans. Given the study focuses on Sumatran oranges, it could be useful to also cite here: <https://link.springer.com/article/10.1007/s00265-012-1463-8>

** This reference has now been included.

161-163: There seems to be here an interchange between “patches” and “food items”, but these are not necessarily equivalent given that the same patch (tree/sps) can provide more than one food item (e.g., leaves, young leaves, seed, endocarp, et cetera). This relates to my request for clarification above. I don't mean to be pedantic here: I am confident that using 27 or any similar adjusted number will not qualitatively affect the analyses and results, and I absolutely do not require the authors to re-run analyses using a different number here. A simple rhetorical clarification will do please.

** We see the reviewer's point. Please see our response to the earlier comment raised by Reviewer 1 on this matter. Having one food type in a given patch ensured that our estimates from the wild, and the design of our ABM, were comparable.

...-205: Congratulations to the authors for having been able to assemble the quantity and quality of data necessary to extract these social demographics and behaviours and the ingenuity of building a ABM world inspired on these parameters!

** We thank the reviewer for this acknowledgement.

272: Phrasing could be adjusted here given that 224 barely “surpasses” 223.

** We have changed this to 'reached' rather than 'surpasses'.

389: 'clarity'

** We have changed the sentence to remove 'offer' so it now reads 'may further clarify'.

395: 'these data'

** Changed to 'these'.

430: 'which' -> 'that'

** Changed to 'that'.

451: hominins + great apes = hominids

** We have combined these into 'hominids'. In the sentence before, we keep 'humans and great apes' separated so that readers outside of anthropology can infer the meaning of terms such as 'hominids' if they are not already familiar with them.

469: Aren't there references worth citing here about great ape cultures in non-food repertoires?

** We have added reference to Malherbe et al. 2025 *Current Biology*, which illustrates cultural loss in sets of communicative gestures used by chimpanzees through human disturbance. We have also added a reference to a recently published paper - Permana et al. 2025 *Communications Biology* - illustrating that social learning may guide choice in orangutans construction behaviours (i.e. tree species selected for nest building).

Reviewer #2 (Remarks to the Author):

Line numbers correspond to the tracked-changes version of our manuscript.

This study introduces an agent-based model designed to test the effects of learning on the breadth of the diet repertoire. In the model learning is understood as happening in social situations of graded intensities, which facilitate the learning process and accelerate learning about handling and consumption of newly encountered food items. The observation part of the learning process is followed by independent exploration of the respective food item. After a certain number of explorations the new item is added to the diet repertoire. Independent explorations thus serve the purpose of consolidation of the learning process.

I will focus in my comments on design and training of the agent-based model, as well as the design of the experiments which were carried out by the model. The model as such is carefully designed and clearly presented in the manuscript, the method section as well as the supplementary documents provided.

** We thank the reviewer for these comments, and are glad to hear that our models' design and outputs are communicated clearly.

Model design and training:

The environment in the model consists of patches, which contain a single type of food item and may be occupied by other individuals with whom the agent can interact. Although each of the food items is randomly selected in the frequency in which it appears in the training data, the items as such do not cluster seasonally. Since the model is not designed to be spatially explicit, this may be considered as negligible. There are also not episodes of shortage in food items included. Yet, future versions of the model certainly benefit from a more realistic and varying presentation of food items.

** All statements here are correct. We provide several recommendations for how some of these factors - e.g. seasonality of foods, and periods of scarcity - should be pursued by further research (see Supplementary Information), but agree with the reviewer that they are likely negligible for our model design, as they are implicit in food encounter probability estimations.

Crucial factors in the model were collected from an extensive dataset collected in Suaq Balimbing in the Gunung Leuser National Park in Sumatra. Technically, these factors constitute the initialization parameters in the modeling process. In this model, this includes for instance the frequency of the food items, age-dependent frequency of encounters with conspecifics as well as social state assigned, probability of exploration, number of explorations required to include food items into the diet repertoire, and possibly more. Although all of the factors are explained in the manuscript and/or method section I would like to suggest two corrections, which would certainly increase the readability of the manuscript. Because initializing parameters are crucial for the model outcome, the authors may consider listing them in a separate table. This would also clarify the differences among the three scenarios which were further examined in this study. Moreover, the statistics may be shifted to this table. At present they are presented in brackets following the respective factor state in the main text. Shifting them to a separate table would certainly improve readability.

** All of this information is currently available in the tables we present in the Supplementary Information (see Tables S1-S5), including in the format requested by Reviewer 2. Given the extensive number of models we have used to calculate estimations from wild data, tabulating these results produced four distinct tables (Tables S2-S5), plus an additional table for our ABM's conditions (Table S1). We therefore do not have space to move these tables into the main manuscript, and believe that doing so would likely saturate readers with information, and dilute the readability of our paper. However, we have signposted the availability of these tables in the SI more explicitly in the main text. At each instance where we explain which estimates from real-world data are translated into ABM coefficients, we signpost Table S1 (see examples on lines 303, 312, & 352, as well as throughout the section '*Exploration in each social state*'), as well as in the legend of Figure 3, which illustrates the trends found in data from wild orangutans. We also signpost the availability of Tables S2-5 in the methods (see examples in section '*Social states across development*').

Design of the experiments:

Three scenarios were examined. Exposure, Enhancement & Peering is the most inclusive one, and the one which is based on the training data set. Both of the other ones, Exposure & Enhancement as well as Exposure only, represent reduced learning processes, which do not occur in reality. Consequently, the performance of both of the 'inferior' learning strategies leads to a smaller dietary repertoire, as shown in figure 4. As far as I understood, the strategy that agents solely explore potential food items on their own is not included in the experimental set-up, because it is considered too dangerous.

** Reviewer 2 is correct that the treatment '*Exposure, Enhancement & Peering*' is the one we consider to be in-keeping with previous research of wild orangutan social learning and foraging (However, as noted, it should not be assumed all instances of peering, or close association lead to learning in every instance - evaluating their long-term effects on learning is a key contribution of our paper). The other simulation treatments involve stepwise removal of forms of social learning, starting with observational learning through *Peering* (as it is generally considered more complex than both '*Exposure* and '*Enhancement*'; this produced the treatment group '*Exposure & Enhancement*'), and then subsequently, we removed both *Peering* and *Enhancement* from our ABM (leaving the treatment group '*Exposure Only*').

It should be noted, however, that the reduced scenarios '*Exposure & Enhancement*' and '*Exposure*' were still informed by substantial real-world data. Because wild immature orangutans enter close association without peering (thus, enter association proximities conducive with '*Enhancement*'), we were still able to estimate the likelihood of immatures entering close proximity and its consequential effects on exploration. Similarly, we could estimate the likelihood of wild immatures exploring at each age when in distant association (> 10 m). In the '*Exposure & Enhancement*' treatment, simulated immatures could enter close and distant association, and their likelihood of exploration was influenced by this association proximity. In the '*Exposure Only*' category, simulated immatures could only enter distant association.

The use of real-world data for these scenarios is important because, whilst we can observe how close and distant association influence short-term exploration in the wild, we cannot know how this translates into long-term development (both with and without peering). This is the key question our simulation aims to answer.

The reviewer is correct that we cannot remove the effects of *Exposure* on diet development in a data-driven way, because dependent immatures are in near constant association with their mothers. Resultantly, we do not have available data on many feeding patches that dependent immatures would identify and visit if they were not led there by their mothers. Maternal association is vital for wild immatures across the dependency period, up until the onset of independence (7-9 years) and previous studies suggest that this is when the majority of diet learning takes place.

To get an initial characterization of how reduced *Exposure* may influence diet development (once *Enhancement* and *Peering* were removed from the ABM), we ran simulations similar to the '*Exposure Only*' treatment, but with stepwise reductions in the mean number of food items simulated immatures visited per day (In the '*Exposure Only*' treatment, simulated immatures encountered, on average, 27 patches/day, and this was reduced to 25, 15, 10, and 5 for further simulations; N = 250 iterations each). The results for this can be seen in Extended Data Figure 1; however we emphasize in the figure legend that these simulations are speculative, and without the vital role of *Exposure* (as well as other Maternally-provided behaviours), dependent immatures are extremely unlikely to survive, let alone expand their diets. We therefore use *Exposure* as a critical baseline for diet learning, in a context which makes sense to wild orangutans, and then quantify the relative contributions of other forms of social learning (*Enhancement & Peering*) on top of this baseline.

I gather from the description of the strategies, that the agents keep a particular strategy throughout the entire duration of the experiment and they are not shifting for instance from Exposure, Enhancement & Peering until onset of independence to Exposure & Enhancement until the end of the experiment. In this sense, the scenarios are designed uniform and consistent. It would nevertheless be interesting to examine the impact of a change in strategies in the course of the experiments.

** As the reviewer suggests, these strategies are consistent throughout the entire simulated timeframe. I.e. an individual in the '*Exposure, Enhancement and Peering*' treatment can have their exploration enhanced by either *Enhancement* or *Peering* across the entire simulation. The probability of either social state enhancing learning is age-dependent, and involves:

[1] A hierarchical decision about the social state (based on the age-dependent probability of being in association with a conspecific, then whether this is a close association, and then whether the immature peers when in close association).

[2] A decision as to whether the simulated immature explores the encountered food (based on the social state and age). Importantly, for each age and social state, we calculated the likelihood of simulated immatures exploring based on whether wild immatures explored in similar contexts (see section *Exploration in each social state*). This means, much like in wild individuals, although close association or peering enhance the likelihood of learning, being in a social state does not necessarily have to lead to learning: i.e. a simulated immature may end up in a *Peering* state, but fail to explore. This probability of subsequent exploration and learning is therefore an additional stochastic factor in our model which simulates real life.

Individuals in the '*Exposure & Enhancement*' category could benefit from *Enhancement* at any age; however importantly, they could not peer at any age. Individuals in the '*Exposure Only*' category had no access to *Enhancement* or *Peering*.

Whilst we could investigate how swapping strategies at different ages could influence our results (e.g. prohibiting *Peering* during the dependency phase, and then enabling *Peering* following independence), we do not think this will bring us closer to our intended research questions. Introducing arbitrary choices about when new

forms of social learning become available to immatures during development would dilute one of the core strengths of our paper: that we make design choices from our ABM that are driven exclusively by what can be observed in wild individuals (which we can subsequently validate). Moreover, by removing forms of social learning from the very beginning of the simulation - and keeping these conditions constant across the simulation - we can quantify the importance of social learning over entire developmental trajectories. Adding additional simulations suggested by the reviewer would likely confuse readers, and would not have immediate biological relevance. For these reasons, we refrain from including these suggestions.

We do, however, believe that simulations similar to those that the reviewer suggests could be useful for future studies. For example, if an immature orangutan is taken into rehabilitation (following being orphaned), one could ask whether they are still able to learn their full diet repertoire after being released at a later age, assuming there is no formal teaching about food items during development (see example simulation below). We touch on the possible importance of diet teaching during rehabilitation in our manuscript (see Discussion), however, we do not currently have the available data to ensure that these simulations are constructed accurately. Thus, we leave this for future work when the data becomes available.

As the reviewer can likely understand, part of the beauty of our ABM is that it has been carefully constructed to rely exclusively on real-world inputs, for which it produces similar outputs to real-world phenomena. We believe that using a limited, yet highly-valid parameter space to run our model will be looked upon favourably by readers of our paper, given that it ensures our model is valid, informative, and readily comprehensible (positive appraisals which are echoed by Reviewers 1 & 3). For these reasons, we will keep the simulated treatments as they are, and hope to investigate the possibility of further ABMs which tackle questions similar to what Reviewer 2 has suggested if and when the required data becomes available.

Example simulation

We provide an illustration of how rates of peering and enhancement within our ABM *could* be modified – including preventing different types of learning at different stages – to answer further questions as the reviewer suggests. However, these simulations are outside of the scope of our manuscript, as required data is not available (See below). However, these simulations can form the basis of further work following publication if the necessary data for ABM calibration can be acquired.

This figure illustrates a hypothetical example where immature orangutans are orphaned after living for 2 years (red development curve; before this age, individuals can enter close association, and peer). From 2 years, each individual enters rehabilitation; however, in this scenario, the rehabilitation efforts do not involve teaching orangutans about the food items that occur naturally within their own habitats (thus, probability of learning new foods is 0 following 2 years). Following release back into the habitat at 7 years, we can examine how the diet

repertoires change, assuming that rehabilitant's social and exploratory behaviours are similar to individuals of the same age class.

This hypothetical scenario gives an initial insight into whether rates of peering and social enhancement at older ages (i.e. across the independent phase) can act to rectify reduced diet learning across the dependency phase – initial results suggest that they cannot. This would suggest that teaching immature orangutans about ecologically-available food is vital during rehabilitation (further supporting the case that rehabilitation centres should continue to do so). However, further data would be needed to answer these questions robustly, including information about: [1] the rates of independent exploration of rehabilitants following release (see discussion in our manuscript), and [2] whether sociability of rehabilitant orangutans reflects those of wild individuals. Thus, answering this question requires more data, and falls outside of the scope of our current study.

Discussion:

In view of the results presented in figure 4, the authors claim in lines 305 and 306 that they provided “the first to robustly evidence that these effects [of social learning] exist within the culture of a non-human species”. Actually, I consider this quite a bold claim, given that the ABM was trained with an enormous data set, and the agents were explicitly designed to act that way. If the results of the experiments would not have matched the training data set, the ABM would have been designed the wrong way. I do not wish to doubt that social learning is crucial to acquire an extensive diet repertoire, but the way the authors phrase it here, implies that they may have confused cause and effect. The ABM was designed to reproduce the effects of social learning. The results of the Exposure, Enhancement & Peering scenario therefore should reproduce learning success in the training data. Otherwise, the ABM would have been wrongly designed.

** There are a number of reasons why our ABM is not limited by these points raised by the reviewer:

1. We do not enter into circular reasoning, as our ABM looks to test whether simple, long-term exposure is enough for an orangutans' diet to develop. This is by no means a trivial question. Decades of research have struggled to parse apart whether social learning permits wild apes to acquire information which they could not otherwise learn through more limited (or no) social interactions. Given that apes encounter hundreds of thousands of opportunities for exploration during immaturity (in the case of our ABM, >148,000 learning opportunities), the null hypothesis is that all of apes' cultural knowledge could be acquired through exposure and independent exploration only - regardless of the influence of other forms of social learning - has until this point been an argument defying empirical testing. In the case of diet breadth, this would suggest that having hundreds-of-thousands of opportunities to explore various food items would be enough to learn an adult-like diet, without the influence of enhancement and peering on diet development. Therefore, if this null hypothesis was true, removing the higher rates of exploration offered by *Enhancement* and *Peering* from our ABM would have little effect on long-term diet development, as lower-levels of exploration across this enormous number of learning opportunities would be enough to produce an adult-like diet.

We show that this is not the case. We are able to make this point by using a bottom-up approach to estimating wild orangutans short-term behaviours, and seeing how these behaviours interact over extended development (see bullet 2.). Importantly, the additional effects of *Peering* and *Enhancement* on long-term diet learning could have been nonexistent. This was to be expected if the null hypothesis were true: that massive, long-term exposure can still facilitate diet development. Our model provides critical evidence against this claim.

We have clarified this in the manuscript by [1] making the null explanation for diet development (that *Exposure* alone is sufficient) more explicit in our introduction (see lines 93-100 & 155-158), and [2] also through more careful phrasing in our Discussion (see lines 679-689).

2. Reviewer 2 elaborates on the proposed circularity problem, focussing on how we use extensive data to train an ABM of diet repertoire development, and this extensive training 'should reproduce the effects of diet development'. This does not follow. To be clear, whilst we aimed to capture all key aspects of diet development in our model, all aspects of our ABM were calibrated from the 'bottom-up', i.e. we calibrated agent's behaviours only using localized behaviours observed in the wild (e.g. peering, exploring, being in close or distant association, the types of food encountered, etc.). Absolutely no aspects of our model 'training' involved top-down coercion of how diets should develop (e.g. having parameters which control the maximum rate of growth at different ages). This means that the only reason why diet development matched what is observed in the wild is because we carefully selected the relevant behaviours, and estimated their performance across development with exceptional accuracy. The long-term effects of these behaviours could then emerge spontaneously during the process of our ABM. Notably, we did not have to tweak any of the model parameters for the model to predict diet repertoire development accurately, which underlines how well we could estimate these parameters with the help of the extensive long-term data. As mentioned above, we were then able to test whether adult-

like diets could emerge through extensive exposure and low exploration rates alone, which is a fundamental research question in our field.

3. The reviewer elaborates further, and states that if the model treatment that included all forms of social learning did not reproduce the outcomes observed in the wild, it would be considered to be “wrong”. This is not the case. As mentioned, careful model design has allowed us to reach the gold standard of ABM validation (‘output validation’) where the outputs of the model match those observed in real life (i.e. the timings of adult-like diet development). However, if this validation was not achieved, there are many other forms of model validation which could have been used to assess the real-world validity of our ABM, including:

- distribution validation (i.e. the mean timings of diet development significantly different between simulated and wild environments, but their general distribution is similar in both environments);
- ‘pattern’ validation (i.e. does diet development follow a general pattern which is similar to those of wild individuals).

Both of these forms of validation are generally considered to be weaker than output validation. Thus we focussed on output validation to check the real-world relevance of our model. Moreover, as we use long term cross-sectional data for model calibration, pattern validation was not possible across the simulated developmental period.

Importantly, the existence of other forms of model validation mean that we did not necessarily need to achieve output validation to have a ‘right’ or ‘wrong’ model (thus, countering the reviewer’s point about circular reasoning). For example, if we did not achieve adult-like diets by the onset of independence when all forms of social learning were included in our model, it would still be possible to take the simulated age that the ABM predicted that adult-like diets would arise (which could be subject to a level of distribution validation), and then to assess whether the removal of forms of social learning led to diet development which deviated from this simulated benchmark. Theoretically, even for this rescaled model, any change in diet development caused by the removal of social learning (e.g. 20% smaller diets by the same time point), could then be reverse translated back into a prediction for how social learning influences diet development based on the timings observed in the wild. Therefore, whilst having output validation allowed us to quantify the effects of social learning more directly, a situation where our model did not achieve output validation would still have been useful for answering our research question.

4. In addition to being able to test whether social learning is necessary for diet development, we were able to use our ABM to quantify the relative contribution of different forms of social learning by removing them from the model in a stepwise fashion. We began by removing *Peering* from our ABM, as aspects of observational social learning are hypothesized to be particularly important for specific aspects of diet development, such as learning about difficult-to-process or rare foods. We then removed *Enhancement* in addition to *Peering*. Our results provide critical insight into the relative contribution of each form of social learning: namely, that *Enhancement* and *Exposure* are responsible for learning the majority of orangutans’ adult diets, and that *Peering* contributes to the development of a smaller yet critical additional proportion of the growing diet (as previously hypothesized). Quantifying the contributions of different forms of social learning to long-term, broad-scale development is impossible in the wild; thus our study outlines the first methodology to tackle this quantification in a systematic and precise way. Our study therefore greatly enhances our knowledge of how different forms of cultural inheritance shape orangutans’ development. We have modified our introduction to highlight this aim more explicitly (see lines 158-160) and we discuss this key implication in the third paragraph of our discussion.

We believe that these points clarify all concerns raised by the reviewer. The claims we make about the advantages of our ABM’s design, and the validity of its results, remain wholly intact. However, using these valuable points arising from the reviewer’s comments, we have been able to further clarify sections of text in the main manuscript, ensuring that our results are transparently communicated and logically justified. We also now also include further information about our choices for model validation within the Supplementary Information (see end of section 2.2, ‘*A note on ABM validation*’).

In lines 313 and 314, the authors state that the “removal of even one form of social learning (...) significantly decelerates diet repertoire development”. In view of Figure 4, I do not see significant differences particularly in the early sections of Exposure, Enhancement & Peering and Exposure & Enhancement. In fact, both curves are widely overlapping as well as the confidence intervals until the age of 6. In the model, this is the episode, in which the major body of knowledge on the diet repertoire is acquired in the model. It is only the expansion of the diet repertoire in the Exposure only scenario, which significantly deviates from both of the others. Additionally, the effects displayed in figure 4 illustrate the increase in the diet repertoire. Inferior strategies lead to a smaller dietary repertoire, but not necessarily to a deceleration in the development. Rephrasing these lines may help to better understand the results shown in figure 4.

** We understand the point made here by Reviewer 2. We used statistical models to compare the diet-repertoire size at both 9 years (onset of independence) and at the end of immaturity (15 years) between the different treatments. Contrary to what Reviewer 2 suggests, we use these models to report a significant difference in the size of diets by these specific time points, and given that these time points are fixed across all experimental treatments (i.e. maximum age at onset of independence is always taken to be 9 years), it is only logical to infer that the average rate of diet development varies between treatments (where average rate of diet development = total diet size/total time elapsed).

However, we agree that we failed to properly acknowledge that initial diet development in the experimental treatments '*Exposure, Enhancement & Peering*' and '*Exposure & Enhancement*' are similar across initial years of the model, and without the inclusion of *Peering*, diet development slows later on during the dependency phase. This suggests that the removal of *Peering* on the rate of diet-repertoire development becomes most pronounced at later ages of dependency. We have modified the text to distinguish this result more clearly, and we believe that including this result offers a more precise interpretation of how social learning influences the rate of diet development within our ABM. See lines 463-468.

Besides, I do not understand, where the angles in the curve for the Exposure only scenario may come from (around age 1 and age 3). Is there an explanation for this effect?

** This effect can be traced to the exploration rates when immatures are outside of 10m across development (see Fig. 3d). Exploration rates when immatures were in distant association was low prior to 1 years old. Following 1 year of age, exploration rates in distant association increased, which means that diet learning will accelerate; however following 2-4 years, exploration rates in distant association once again decreased, thus slowing diet development again. Given that in the '*Exposure Only*' treatment, the exploration rates in distant association are the only determinants of whether an individual would explore a food item (i.e. there is no *Enhancement* or *Peering*), it makes sense for diet development to trace this relationship in exploration rates closely. We now clarify this relationship in the Results, see lines 466-468.

Minor corrections:

line 687 - typo: 3 = 3 explorations

** These have been corrected. The number of explorations for complexity categories have also been corrected to what is reflected in the text.

In sum, the authors introduce in this study a thought-provoking ABM, which will certainly inspire further experiments.

** We thank the reviewer for their comments and helpful suggestions!

Reviewer #3 (Remarks to the Author):

Line numbers correspond to the tracked-changes version of our manuscript.

Dear Elliot Howard-Spink, Caroline Schuppli, and Co-Authors,

Thank you for submitting your manuscript "Culture is critical in driving orangutan diet development past individual potentials" to *Nature Human Behaviour*. I thoroughly enjoyed reviewing your work, which I find to be an innovative and impactful contribution to primatology and cultural evolution. Your use of an empirically calibrated agent-based model (ABM) to demonstrate the necessity of social learning for orangutan diet repertoires is rigorous and novel, providing the first robust evidence of culturally dependent repertoires in a non-human species. The study's implications for human evolutionary origins and orangutan conservation are compelling.

** We thank the reviewer for the positive assessment!

Outlined below to enhance clarity and address a few points:

1. The abstract could better emphasize your key finding on culturally dependent repertoires. Consider revising to highlight the novelty.

** We have now highlighted the key finding of orangutan diets being culturally-dependent repertoires in the abstract.

2. Lines 17-44: The shift from human to animal culture is abrupt. A bridging sentence (e.g., "While human culture exhibits cumulative evolution, simpler cultural dependence may characterize other species") could improve flow.

** Given the strict word limit, we have not been able to add an extra sentence here (and Reviewer 1 recommended we remove our initial bridging sentence); however we do state that animal models are useful for understanding human cultural evolution in our abstract, thus the reader should be primed to expect discussion of both humans and animals in our introduction.

3. Lines 294-296: Figure 4a's "Zone of No Exposure" is visually compelling but unexplained in the Results. A sentence linking it to Extended Data Fig. 1 (line 897) would clarify its relevance.

** We now provide two sentences at the end of the Results clarifying the relevance of the 'Zone of No Exposure', and signpost Extended Data Figure 1 for tentative internal resolution of the developmental trajectories within this zone.

4. Lines 398-411: Your justification for not removing exposure is sound, but framing it as an opportunity for future experimental work (e.g., captive studies) could strengthen this section.

** In this paragraph, we cite some initial work on captive orangutans which suggests that exposure is an important social learning mechanism for diet development (Hardus et al. 2015 *Primates*). We are hesitant to provide more specific suggestions for further research on the effects of exposure on diet repertoire development, as reducing the diversity of foods provided to captive-living primates may spark ethical concerns. Whilst the cited paper addresses these issues with care, it is unlikely that we have sufficient available space in the current paper to provide specific suggestions for further research in a way which is sensitive to captive individuals' wellbeing. We therefore suggest that further study of the importance of exposure will likely be a useful line of future research, and give this study as an example. (See Lines 833-836).

5. Lines 447-453: The suggestion of a shared capacity with the human-ape ancestor is exciting but speculative. Toning it down (e.g., "suggests a potential shared capacity" or "may reflect an ancestral trait") or citing supporting evidence could refine this.

** We have adjusted these claims in a similar style suggested by the reviewer.

6. Lines 626-627 set alone exploration equal to distant association due to data gaps. This assumption's impact on results (e.g., underestimating independent learning) could be briefly noted.

** We agree and now touch on this point in our Supplementary Information, in the section titled: "We set the exploration rate when simulated immatures are 'alone' to be equal to when they are in 'distant association'."

As immatures are virtually never alone during the dependency period, we could not directly estimate exploration rates when wild dependent immatures were not associated with other individuals. We therefore set their exploration rates when alone to be the same as when in 'distant association' at each age (between 10-50 m away from every other conspecific). Previous research on wild orangutans of older age classes (i.e. adults) has revealed that being in association leads to slightly higher exploration rates than when alone (Schuppli et al. 2017). Thus, by equating exploration rates when in distant association and alone, our model may somewhat overestimate the likelihood of exploration when immatures are by themselves. Ultimately, this would make our model's baseline condition (*'Exposure Only'*) a conservative estimate of diet development without peering and enhancement (i.e. without social learning, lower exploration rates when alone in real life would lead to slower diet development). This further emphasizes our conclusion that social learning is necessary to accelerate diet learning to meet orangutans' developmental milestones.

7. Lines 682-687 extrapolate exploration needs for rare foods (3-5 complexity). Acknowledging potential errors (e.g., "may underestimate rare item learning") would enhance rigour.

** This is a valuable point raised by the reviewer, as it is one of the only instances where we had to extrapolate data to estimate behaviours of wild individuals. We now discuss the possible assumptions made by our extrapolation, and its effects on our results, within the Supplementary Information (see section *'The number of required explorations for more complex foods (complexity level 3 and above) are estimated via extrapolation.'*).

As the reviewer suggests, we agree that this linear extrapolation may potentially underestimate the importance of social learning for acquiring rare and complex food items. This strengthens the conclusions of our paper. We designed our ABM to evaluate the long-term, broad-scale importance of social learning for diet development, and where possible, have aimed to be conservative when specifying the effects of social learning on behaviours that cannot be directly estimated from wild data. This means that individuals who are in the condition where there is no *Peering* nor *Enhancement* still have a reasonable ability to incorporate complex and rare foods in their diets (which, in real life, may be harder to learn than our linear trajectory predicts, particularly without observational social learning). Yet, even with these conservative assumptions, we demonstrate that it is not possible to produce similarly sized diets without social learning. We therefore agree with the reviewer that this is a key aspect of our model design to raise in our Supplementary Information, and also believe that the central message of our paper is even stronger with this noted consideration.

8. Figure 1's captions could clarify how peering/exploration links to diet learning (e.g., "Peering (a) facilitates targeted exploration (b), accelerating food acquisition").

** We have added a sentence on how peering guides object exploration and ultimately diet learning.

9. Figure 4's "Zone of No Exposure" is intriguing—consider briefly mentioning it in the main text to tie it in.

** We have now included a description for the Zone of No Exposure at the end of the Results section, and also we discuss its relevance to our study in the Discussion (see lines 481-487).

10. Minor Edits: Check "nutrient" (line 62) to "nutrients," "adult-like repertoires" (line 154) to singular for consistency, and "class of association" (line 211) for completion (perhaps "close association").

** Nutrient is kept the same as it is an adjective for 'demands'. However we have changed 'demands' to 'requirements' as we believe the confusion comes from whether 'demands' is a verb or noun here. We have implemented all other changes.

Your methodology is exemplary, and the validation against wild data is a standout feature. These minor revisions aim to maximize accessibility and impact for Nature Human Behaviour's broad audience. I congratulate you on this excellent work and look forward to seeing it published.

** We thank the reviewer for their comments and positive appraisal.

Reviewer #3 (Remarks on code availability):

The authors provide a view-only link to their data and code on the Open Science Framework (OSF) at https://osf.io/wndx8/?view_only=dbfa236e546343b7a5994117dc879acc. While the link is functional (accessed March 31, 2025, 09:20 AM PDT), the embargo noted in the manuscript (until publication) prevented me from fully downloading and testing the code. The OSF page includes a README file and

structured directories, suggesting a commitment to transparency. However, without full access, I cannot confirm the code's completeness or usability (e.g., installation instructions, dependencies, or runtime functionality). The README outlines the ABM's structure and calibration process, which is promising, but its sufficiency for community use remains unverified pre-embargo.

Assuming the embargo lifts upon publication, as stated, the code's potential for reproducibility seems high, given the detailed Methods section and validation against wild data. However, I recommend the authors clarify post-publication access details (e.g., updating to a public DOI) and ensure the README provides explicit instructions for installation and execution. This would maximize the study's impact and utility for the research community.

****We have ensured that all data and code in the OSF archive can be downloaded - we apologise that this could not be accessed previously. All information about versions of packages can be found in our supplementary materials, ensuring that our results are fully replicable.**

Post publication, all code and data will be available at Mendeley Data (in a repository with associated DOI). This DOI has been reserved (10.17632/7kvr22vk5f.3), and we will activate this link upon acceptance, to archive the data and code associated with the accepted version of our manuscript for perpetuity.

As requested by Reviewer 3, we have also included all instructions of how to install R & Python (including links to where they can be freely downloaded), as well as instructions on how to import required packages and modules, and how to load these dependencies into the R and Python environments.